# Kalman-Based Joint Analysis of IMU and Plantar-Pressure Data During Speed-Skating Slideboard Training

**DOI:** 10.3390/s26010272

**Published:** 2026-01-01

**Authors:** Huan Wang, Luye Zong, Guodong Ma, Keqiang Zong

**Affiliations:** 1School of Ice and Snow Sports, Jilin Sport University, 5699 Linhe Street, Economic and Technological Development Zone, Changchun 130022, China; huanwang705@gmail.com (H.W.); luye84944@gmail.com (L.Z.); taozitaotaozi@gmail.com (G.M.); 2School of Physical Education, Zaozhuang University, No. 1 Minsheng Road, Xuecheng District, Zaozhuang 277100, China

**Keywords:** IMU, plantar pressure, speed skating, slideboard, Kalman filtering, biomechanics, movement analysis

## Abstract

Efficient monitoring of lower-limb coordination is important for understanding movement characteristics during off-ice speed-skating training. This study aimed to develop an analytical framework to characterize the kinematic–kinetic coupling of the lower limbs during slideboard skating tasks using wearable sensors. Eight national-level junior speed skaters performed standardized simulated skating movements on a slideboard while wearing sixteen six-axis inertial measurement units (IMUs) and Pedar-X in-shoe plantar-pressure insoles. Joint-angle trajectories and plantar-pressure signals were temporally synchronized and preprocessed using a Kalman-based multimodal state-estimation approach. Third-order polynomial regression models were applied to examine the nonlinear relationships between hip–knee joint angles and plantar loading across four distinct movement phases. The results demonstrated consistent coupling patterns between angular displacement and peak plantar pressure across phases (R^2^ = 0.72–0.84, *p* < 0.01), indicating coordinated behavior between joint kinematics and plantar kinetics during simulated skating movements. These findings demonstrate the feasibility of a Kalman-based joint analysis framework for fine-grained assessment of lower-limb coordination in slideboard speed-skating training and provide a methodological basis for future investigations using wearable sensor systems.

## 1. Introduction

The efficient monitoring of human motion is important for understanding movement characteristics and supporting training analysis in high-intensity sports. In speed skating, the technical execution of the push-off and recovery phases plays a key role in lower-limb coordination and movement control. However, assessing these biomechanical features outside the ice rink remains challenging due to the absence of accurate feedback mechanisms and field-deployable systems. The slideboard, a widely used dryland training tool, simulates the lateral gliding and push-off patterns of on-ice skating and provides an effective environment for neuromuscular and strength conditioning [1]. Despite its popularity, the quantitative evaluation of slideboard training performance remains underdeveloped.

In recent years, wearable sensors, such as inertial measurement units (IMUs) and plantar-pressure insoles, have become widely adopted in sports biomechanics for capturing kinematic and kinetic data in real-world environments [2,3]. IMUs can measure multi-axis acceleration and angular velocity, providing continuous estimation of joint motion during dynamic activities [4]. Meanwhile, pressure insole systems quantify plantar load distribution, reflecting the athlete’s balance and force application strategies [5]. The integration of these two sensor modalities enables a comprehensive understanding of lower-limb coordination during movement, bridging the gap between laboratory biomechanics and field-based monitoring [6,7].

Sensor fusion has emerged as a promising approach for enhancing motion analysis accuracy and interpretability [8]. Previous studies have demonstrated that combining IMU and pressure data improves gait phase detection and dynamic stability assessment in walking and running tasks [9,10]. However, limited research has explored the joint analysis of kinematic and plantar-pressure signals in skating-related training. Speed skating is characterized by asymmetrical lateral propulsion, large joint excursions, and complex coordination between hip, knee, and ankle joints [11]. Understanding the kinematic-kinetic coupling of these joints may contribute to the characterization of movement organization and provide informative descriptors for training analysis [12]. Therefore, developing a wearable, Kalman-based multimodal state-estimation framework for the joint analysis of kinematic and plantar-pressure data during slideboard training has scientific relevance.

This study aimed to develop a Kalman-based multimodal state-estimation approach for the joint analysis of IMU-derived kinematic data and plantar-pressure signals to characterize lower-limb coordination during speed-skating slideboard training. Specifically, this study examined the nonlinear relationships between joint-angle variations and plantar loading across distinct motion phases. It was hypothesized that the joint estimation of IMU kinematics and plantar-pressure signals would reveal consistent coupling patterns between kinematics and kinetics, thereby demonstrating the feasibility of wearable sensor-based approaches for fine-grained analysis of lower-limb coordination in speed-skating dryland exercises.

## 2. Proposed Method

Eight elite male youth speed skaters from China voluntarily participate in this study, including three national-level master athletes and five first-level athletes certified by the General Administration of Sport of China. The participants have an average age of 16.38 ± 0.92 years (range: 15–18 years), height of 178.13 ± 6.03 cm, body mass of 65.04 ± 4.85 kg, skeletal muscle mass of 33.04 ± 2.52 kg, body fat percentage of 9.23 ± 2.11%, and an average training experience of 6.0 ± 0.76 years (range: 5–7 years). The basic characteristics of the participants are presented in Table 1.

All participants undergo systematic on-ice training within the three months preceding the experiment and report no musculoskeletal injuries or chronic diseases. Each participant is fully informed of the research objectives and procedures and provides written informed consent prior to participation.

### 2.1. Instruments

#### 2.1.1. Wearable IMU Motion-Capture System (MyoMotion)

All kinematic data in this study are collected using the wearable inertial measurement unit (IMU) motion-capture system, MyoMotion (Noraxon U.S.A. Inc., Scottsdale, AZ, USA). The MyoMotion system is a high-precision three-dimensional motion-capture device developed by Noraxon, designed for quantitative movement analysis, clinical gait assessment, rehabilitation monitoring, and ergonomics research.

The system comprises 16 IMU sensors (Research Pro IMU Sensors) and data processing software (MR 3.14). Each IMU sensor weighs approximately 37 g and measures 52.2 mm × 37.8 mm × 18.1 mm. Each unit integrates a triaxial accelerometer (measurement range: ±16 g), a triaxial gyroscope (±2000°/s), and a magnetometer (±1.9 G). Sensors are securely attached to specific body segments of the athlete to record the three-dimensional joint angles and linear accelerations during slideboard movements.

The sampling frequency is set to 100 Hz. The system provides real-time 3D motion data with an accuracy of ±1° in static conditions and ±2° in dynamic conditions. MyoMotion allows for synchronous acquisition with other devices such as electromyography (EMG) systems, plantar-pressure platforms, 3D force plates, 2D video cameras, and other third-party biomechanical measurement systems, ensuring comprehensive multimodal motion analysis.

#### 2.1.2. Plantar-Pressure Measurement System (Novel™ Pedar-X)

Plantar-pressure data are collected using the Novel™ Pedar-X system (Novel GmbH, Munich, Germany), a high-precision in-shoe pressure measurement system designed to quantify plantar-pressure distribution and ground reaction forces during various locomotor activities, such as walking, running, and skating [13].

The Pedar-X system consists of capacitive insole pressure sensors, a data acquisition unit, and a wireless transmission interface. Each insole contains 99 capacitive pressure sensors, which transmit data to the computer via Bluetooth wireless communication. The system provides accurate measurements of real-time plantar-pressure values and regional force distribution, making it suitable for dynamic biomechanical analysis.

For analytical purposes, each foot is divided into eight anatomical regions: medial toe, lateral toe, medial forefoot, lateral forefoot, medial midfoot, lateral midfoot, medial heel, and lateral heel. Plantar-pressure intensity in each sensor is categorized into eight levels of pressure magnitude, representing graded pressure intervals. The regional segmentation and pressure classification are illustrated in Figure 1.

Although the total mass of the equipment is less than 1 kg, the present study implements strict controls to minimize any potential influence on plantar-pressure measurements or movement strategies. A preliminary pilot test conducted prior to the formal experiment indicates that the Pedar-X system exerts negligible effects on technical performance. The device is mounted on the lower back rather than on distal lower-limb segments, thereby reducing any direct mechanical impact on plantar loading to the greatest extent possible. Additionally, an acclimation session is provided before data collection to ensure that participants confirm no alterations in movement sensation or technical execution caused by the equipment. Previous studies have likewise demonstrated that the Pedar-X system does not modify plantar-pressure patterns or lower-limb kinematics during dynamic activities.

### 2.2. Data Acquisition

All experimental tests are conducted in a standardized sports biomechanics laboratory with a rectangular testing area measuring 10 m × 8 m. The ambient temperature during data collection is maintained between 20 and 23 °C. All athletes wear tight-fitting training suits and athletic shoes, and they are instructed to avoid high-intensity exercise within 24 h prior to testing to minimize fatigue effects.

After completing a 20 min sport-specific warm-up, a total of 16 inertial measurement units (IMUs) are securely attached to each participant using adjustable elastic straps. The sensors are positioned as follows: one at the back of the head (over the occipital bone); on the lateral aspect of both upper arms (mid-humerus region); on the lateral aspect of both forearms (anterior radial region near the wrist); on the dorsum of both hands; on the upper back (spinal level T3–T4); on the lower back (spinal level T9–T10); on the lumbosacral region (pelvic level); on the anterior aspect of both thighs (near the patella); on the anterior aspect of both shanks (distal tibial region); and finally, on the dorsal surface of both shoes near the ankle joint.

These placements ensure the accurate tracking of segmental motion and inter-joint coordination across the head, trunk, upper limbs, and lower limbs during slideboard training. The specific sensor positions are illustrated in Figure 2.

In the preliminary pilot testing conducted prior to the formal experiment, this study compares two sensor configurations: a lower-limb-only setup using seven IMUs and a full-body model employing sixteen IMUs. Because accurate computation of hip joint angles requires relative orientation information between the trunk and thigh segments, the seven IMU configuration is unable to capture complete hip flexion and abduction angles. Consequently, the full-body sixteen IMUs configuration is adopted. Moreover, the full-body model allows the generation of complete motion-capture videos, providing more comprehensive references for movement-state interpretation and technical analysis. Although a full-body configuration is used, the analytical focus of this study remains on the lower limbs; therefore, only lower-limb IMU data are extracted during processing and are synchronized with plantar-pressure data for subsequent analyses.

All kinematic and kinetic data are collected at a sampling rate of 100 Hz to ensure that the dynamic characteristics of slideboard speed-skating movements are captured with sufficient temporal resolution. The IMU system is initialized using a static pose and zero-bias calibration to correct sensor drift, and built-in filtering algorithms are applied during data acquisition to further mitigate integration-related drift. Prior to each trial, the plantar-pressure insoles are calibrated by alternately lifting each foot off the ground, while the IMU sensors are calibrated by performing a static initialization in their designated positions on the full-body model. Each athlete completes three full skating cycles, with a rest interval of at least two minutes between cycles to prevent fatigue.

To minimize the influence of inter-individual differences in foot morphology on plantar-pressure measurements, each participant is provided with a pair of pressure insoles matched to their foot size. The laboratory maintains a full range of insole sizes to ensure that each sensing region corresponds accurately to the anatomical areas of the athlete’s plantar surface. All testing is conducted on a slideboard, with participants wearing standardized training shoes and an additional slideboard-specific over-sock to maintain a consistent interface between the foot, the insole, and the shoe. These procedures effectively control for foot-shape variability, ensuring that the recorded plantar-pressure distributions reflect genuine differences in technical execution rather than morphological artifacts.

During formal data collection, each participant completes three consecutive full sliding trials to evaluate the movement repeatability. After applying identical preprocessing procedures to the joint-angle and plantar-pressure data from each trial, the resulting curves are superimposed for comparison. This repeatability also ensures that the dataset used for subsequent regression analyses possesses satisfactory reliability.

### 2.3. Experimental Procedure and Signal Synchronization

The Pedar-X plantar-pressure insoles are inserted into the participants’ athletic shoes, and the signal acquisition unit and wireless transmitter are secured to the participants’ lower back using a specialized belt system. The total equipment load is ≤1 kg, ensuring that it does not interfere with the execution of technical movements (Figure 3).

Prior to the formal testing sessions, participants wear protective overshoes specifically designed for slideboard training and stand naturally in the testing area with feet shoulder-width apart. Before each trial, the plantar-pressure insoles are calibrated individually. The calibration procedure requires the participant to stand upright on the slideboard and alternately lift the left and right foot. When a foot is lifted, the plantar-pressure readings for that foot are reset to zero within the system, ensuring that the sensors record a value of zero when the foot is not in contact with the slideboard’s surface. This method enables accurate calibration of the zero-pressure baseline for both feet under unloaded conditions. The calibration process is repeated before every trial to maintain measurement accuracy and to prevent misinterpretation of the data—such as apparent loss of body weight or the impression that the athlete is “floating” during movement.

Following calibration, participants perform the “straight-line sliding” technique on a custom 1.5 m wide slideboard designed for off-ice speed-skating training. Each participant completes two sets of trials, with three repetitions per set. The dataset corresponding to the most complete and technically consistent trial is selected for further analysis.

During data collection, the MyoMotion system’s synchronization function is employed to achieve real-time, wireless synchronization between the 16 IMU sensors and the Pedar-X insoles, both operating at a sampling frequency of 100 Hz. This setup enables simultaneous acquisition of kinematic and biomechanical data, ensuring temporal precision and coherence between the motion-capture and plantar-pressure signals.

### 2.4. Data Processing

#### 2.4.1. Definition of the Gait Cycle

To accurately monitor joint-angle variations and plantar-pressure dynamics during the sliding motion, the gait cycle in this study is defined as beginning at the moment when one foot (e.g., the left) first contacts the slideboard surface and produces an initial plantar-pressure signal and ending after the completion of the full technical movement when that foot finishes the push-off phase and leaves the surface, resulting in the complete disappearance of plantar pressure. Phase segmentation is determined using an integrated multi-source feature approach. The plantar-pressure features are captured as follows: the plantar-pressure curve provides key quantitative markers. The transition from zero pressure to the onset of loading is defined as the initial support phase. The first pressure plateau corresponds to the double-support phase, the first pressure peak represents the single-support phase, and the maximal peak denotes the push-off phase. The joint-angle features are captured as follows: hip, knee, and ankle angle trajectories are analyzed to further confirm the lower-limb movement state corresponding to each phase. Video-based verification is carried out as follows: motion-capture video is used to ensure that the segmented phases are highly consistent with the actual technical execution.

In the slideboard experiment, each participant completes three stride cycles. The first and final cycles include transitional movements—from standing to semi-squat and from semi-squat back to standing—and are therefore excluded from analysis. The middle cycle is the only complete, continuous, and steady-state sliding movement and is selected for further evaluation. Its integrity is verified using synchronized video recordings, IMU data, and plantar-pressure signals to ensure the absence of movement interruptions, signal artifacts, or atypical performance patterns. This selection criterion is uniformly applied to all participants to avoid subjective bias and to ensure that the analyzed data are representative of the true steady-state technique.

This definition ensures that each gait cycle represents one complete sequence of motion—from initial contact to the end of the propulsion phase—capturing the full process of load bearing, support, and push-off during slideboard training. The schematic representation of the gait cycle definition is shown in Figure 4.

#### 2.4.2. Data Sample Selection

This study focuses on the kinematic analysis of lower-limb movements during off-ice speed-skating slideboard training. Specifically, data from seven IMUs are selected from the full-body configuration of sixteen IMUs for analysis. These sensors are positioned at the pelvis and the bilateral hip, knee, and ankle joints to capture the dynamic kinematic characteristics of lower-limb motion.

For each participant, data from one complete gait cycle are selected as representative samples. The lower-limb joint-angle data are synchronized and jointly analyzed with plantar-pressure values and pressure distribution data collected simultaneously.

#### 2.4.3. Joint Smoothing via Multimodal Kalman Estimation

To achieve joint estimation and smoothing of kinematic joint-angle signals acquired from IMU sensors and plantar-pressure signals, a multimodal linear Kalman filter is directly applied to the raw sensor data [14]. All signals are sampled at a frequency of 100 Hz, corresponding to a sampling interval of Δt = 0.01 s.

The state vector is defined to simultaneously include joint angles, plantar pressure, and their first-order time derivatives, resulting in a total of ten state variables, as expressed in Equation (1).(1)$$x_{k}={\left[\begin{array}{c} \theta_{\mathrm{hip}\_\mathrm{flex}}\\ {\dot{\theta }}_{\mathrm{hip}\_\mathrm{flex}}\\ \theta_{\mathrm{hip}\_\mathrm{abd}}\\ {\dot{\theta }}_{\mathrm{hip}\_\mathrm{abd}}\\ \theta_{\mathrm{knee}\_\mathrm{flex}}\\ {\dot{\theta }}_{\mathrm{knee}\_\mathrm{flex}}\\ \theta_{\mathrm{ankle}\_\mathrm{dorsi}}\\ {\dot{\theta }}_{\mathrm{ankle}\_\mathrm{dorsi}}\\ F_{\mathrm{foot}}\\ {\dot{F}}_{\mathrm{foot}} \end{array}\right]}_{k}$$

The system state is assumed to evolve according to a constant-velocity motion model between adjacent sampling intervals. Accordingly, the state transition equation is given in Equation (2).(2)$$x_{k+1}=Ax_{k}+w_{k}$$
where **A** denotes the state transition matrix, and $w_{k}$ represents the process noise, which is assumed to follow a zero-mean Gaussian distribution as defined in Equation (3).(3)$$w_{k}∼N\left(0,Q\right)$$

The state transition matrix **A** is constructed based on the sampling interval **Δ*t*** to describe the linear relationships between the joint angle and plantar pressure states and their first-order derivatives. The process noise covariance matrix **Q** is specified in diagonal form to reflect the dynamic uncertainty associated with different state variables.

The observation vector consists of directly measurable joint angle and plantar pressure signals and is defined as expressed in Equation (4).(4)$$z_{k}=\left[\begin{array}{c} \theta_{hip_{f}lex}\\ \theta_{hip_{a}bd}\\ \theta_{knee_{f}lex}\\ \theta_{ankle_{d}orsi}\\ F_{foot} \end{array}\right]$$

The linear observation model is given by Equation (5).(5)$$z_{k}=Hx_{k}+v_{k}$$
where **H** is a sparse selection matrix that maps the observable variables in the state vector to the measurement space, and $v_{k}$ represents the measurement noise, which is assumed to follow a zero-mean Gaussian distribution as defined in Equation (6).(6)$$v_{k}∼N\left(0,Q\right)$$

The measurement noise covariance matrix **R** is specified in diagonal form, with its elements independently defined according to the noise characteristics of the joint angle and plantar pressure signals.

The initial system state is initialized using the first-frame observation values, and the initial error covariance matrix is set as an identity matrix to reflect initial uncertainty. Within this framework, the system state is recursively estimated at each sampling instant through the standard prediction–update procedure of the linear Kalman filter. This multimodal Kalman filtering approach enables joint smoothing of joint angle and plantar pressure signals without the need for additional low-pass pre-filtering, effectively reducing measurement noise while preserving the true dynamic characteristics of the movement process. All subsequent analyses in this study are based on the outputs of this multimodal Kalman filtering procedure.

### 2.5. Data Analysis Methods

After data collection, all lower-limb joint angles and plantar-pressure signals are exported for unified processing. Descriptive and inferential statistical analyses are performed using IBM SPSS Statistics (version 26.0, IBM Corp., Armonk, NY, USA), and all characteristic data are expressed as mean ± standard deviation (Mean ± SD). For each participant, one representative and complete gait cycle is selected for analysis. The gait cycle is defined based on the temporal profile of plantar pressure: it begins when the foot first contacts the slideboard and produces an initial pressure signal and ends when the same foot completes the support and push-off phases, leaves the slideboard surface, and the plantar-pressure signal returns to zero.

During data preprocessing, the researchers first review three-dimensional motion videos to identify and remove abnormal data points, ensuring accuracy and integrity. Subsequently, IMU and plantar-pressure data are temporally synchronized such that key movement events align precisely across modalities. To attenuate measurement noise, all signals are preprocessed using a multimodal linear Kalman filter. Following preprocessing, Tableau software 2024.1 (Tableau Software, Seattle, WA, USA) is used to generate superimposed plots of joint angles and plantar pressure, providing a visual representation of their dynamic coupling.

For quantitative analysis, third-order polynomial regression models are employed to examine the influence and explanatory contribution of joint angles (independent variables) to plantar pressure (dependent variable). Prior to model fitting, all variables are standardized to remove unit-based discrepancies and improve numerical stability. Each joint angle is modeled against the plantar pressure of the ipsilateral foot to evaluate its independent contribution.

To ensure statistical rigor, several procedures are implemented: 90% CI are computed for key variables to quantify measurement uncertainty; inter-individual variability is assessed using standard deviation and coefficient of variation; measurement errors associated with both IMU and plantar-pressure sensors are estimated, while multimodal Kalman filtering and SD-based procedures are applied to reduce random error; the robustness of the third-order polynomial regression models is evaluated through normality tests of standardized residuals; movement repeatability is examined by comparing key indicators across the three stride cycles completed by each participant.

These procedures collectively ensure the scientific validity, robustness, and reproducibility of the findings.

Furthermore, to investigate the dynamic coupling between lower-limb kinematics and plantar kinetics, this study integrates data from MR 3.14 (Noraxon) and Pedar-X (Novel GmbH, Munich, Germany) within a Kalman-based multimodal state-estimation framework for joint analysis. The Kalman-filtered joint estimates enable the extraction of synchronized patterns between joint-angle trajectories and plantar-pressure variations, thereby establishing quantitative correspondences between kinematic and kinetic parameters.

## 3. Results

### 3.1. Dynamic Characteristics of Lower-Limb Joint Angles and Plantar-Pressure Variations

For each participant, one gait cycle from both the left and right lower limbs was selected for analysis. The corresponding joint-angle and plantar-pressure data were integrated with three-dimensional motion video recordings to obtain a comprehensive understanding of the movement patterns.

The results indicated that during the push-off preparation phase of the supporting leg, the flexion angles of the hip, knee, and ankle joints generally reached their maximum values, reflecting the degree of joint flexion prior to propulsion. As the push-off action progressed and the lower limb gradually extended, these joints reached their maximum extension states, with the corresponding flexion angles decreasing to their minimum values, representing the extent of joint extension during propulsion.

In contrast, the hip abduction angle reached its maximum value at the end of the push-off phase when the lower limb was fully extended, indicating the magnitude of lateral propulsion. The minimum hip abduction angle was observed during the single-leg support phase, suggesting that hip joint motion was reduced during this period to maintain postural stability.

According to the data presented in Table 2, the mean plantar pressures for the left and right feet were 127.66 ± 94.84 and 167.00 ± 125.68 N, respectively. The right foot exhibited a significantly higher plantar-pressure than the left, indicating an asymmetry in push-off force between the two sides.

Paired comparative analyses of plantar pressure and lower-limb joint angles between the left and right sides revealed that the mean plantar pressure of the right foot was significantly higher than that of the left foot (Z = 2.48, *p* = 0.008). Among the joint angle parameters, significant bilateral differences were observed in both hip flexion and hip abduction angles (*p* = 0.008 for both), whereas no significant difference was detected in knee flexion angles between the two sides (Z = 0.35, *p* = 0.727). In addition, although the bilateral difference in ankle dorsiflexion angle did not reach statistical significance (Z = 1.77, *p* = 0.070), a noticeable trend toward asymmetry was observed.

Taken together, these statistical results indicated that during slideboard training, plantar load distribution and selected lower-limb kinematic parameters exhibited significant left–right asymmetries.

Further analyses revealed distinct asymmetries in the angular variations in lower-limb joints between the left and right sides. Specifically, the right hip joint demonstrated a greater range of motion (ROM) in both flexion and abduction, while the right knee joint also exhibited a wider flexion range compared with the left. Moreover, the maximum hip abduction angle, maximum knee flexion angle, and maximum ankle dorsiflexion angle on the right side were all substantially higher than those on the left. In contrast, the hip flexion angles of both sides remained relatively consistent throughout the movement cycle.

A line chart was plotted to visualize the dynamic variations in joint angles and plantar pressure. During slideboard training, the angular trajectories of the lower-limb joints exhibited near-synchronous changes.

### 3.2. Phase-Specific Analysis of Joint Angles and Plantar Pressure

Figure 5 and Figure 6 showed that the time-series patterns of joint angles and plantar pressure were highly synchronized across participants, allowing the slideboard skating cycle to be divided into four phases: initial, double-leg support, single-leg support, and push-off.

(1)Initial Phase

At the onset of slideboard contact, plantar pressure in both feet rose sharply from zero, reflecting rapid load establishment. The lower limbs adopted a deep flexed posture, with hip, knee, and ankle flexion angles of 108.7°, 95.8°, and 23.1° on the left and 115.6°, 95.5°, and 23.7° on the right. Notably, hip abduction showed a pronounced asymmetry (left: −2.4°; right: 12.3°), despite both limbs sharing a similar flexed support strategy. These findings indicated that the initial contact phase was characterized by substantial bilateral loading with asymmetric frontal-plane hip control.

(2)Double-Leg Support Gliding Phase

During double-leg gliding, bilateral plantar pressure continued to increase, with the left foot fluctuating between 90 and 148 N and the right fluctuating between 140 and 193.3 N. Joint angles displayed coordinated adjustments. Left leg: Hip flexion increased to 121.2° and then returned to 111.2°; knee flexion decreased sharply from 95.8° to 66.7°; ankle dorsiflexion was reduced from 23.1° to 15.9°; hip abduction increased from −2.4° to approximately 19°. Right leg: Similar trends were observed, with hip flexion decreasing to 107.6°, knee flexion decreasing to 66.5°, and hip abduction increasing from 12.3° to 21.1°.

These patterns indicated a transition from deep flexion to a more upright posture while simultaneously increasing lateral hip displacement, forming a stable triangular support structure commonly observed in slideboard skating.

(3)Single-Leg Support Phase

As the athlete transitioned to unilateral loading, plantar pressure increased markedly (left: 132.5 → 251.9 N; right: 209.4 → 313.4 N), stabilizing before push-off. Joint kinematics showed pronounced hip and knee flexion with gradual dorsiflexion increases. Left leg: Hip flexion peaked at 120.8°, knee flexion increased to 79.9°, ankle dorsiflexion rose to 22.5°, and hip abduction decreased to −3.0° before rebounding. Right leg: Hip flexion reached 121.8°, knee flexion increased to 83.5°, and ankle dorsiflexion rose to 27.0°, while hip abduction decreased to −1.6° before recovering.

These results demonstrated sustained weight acceptance and center-of-mass lateral shift toward the stance limb, accompanied by reduced hip abduction angles and increased flexion-based stabilization.

(4)Push-Off Phase

The push-off phase was characterized by a rapid rise and fall in plantar pressure (left peak: 298.9 N; right peak: 332.0 N), reflecting explosive propulsive loading followed by unloading near zero. Lower-limb extension occurred sequentially. Left leg: Hip flexion reduced to 89.7°, knee extended from 75.1° to 29.5°, and ankle transitioned from dorsiflexion to plantarflexion (−4.2°). Hip abduction increased sharply to 51.0°. Right leg: Hip flexion declined to 80.8°, knee extended to 30.7°, ankle angle fluctuated before final plantarflexion, and hip abduction rose to 53.3°.

This coordinated hip–knee–ankle extension pattern reflects the primary force-generation mechanism of lateral propulsion, while the large terminal abduction angles indicate strong outward force application during push-off.

### 3.3. Dynamic Characteristics of Plantar-Pressure Distribution

As shown in Figure 7 and Figure 8, the plantar-pressure distribution patterns of the left and right feet exhibited a high degree of similarity. Based on the spatiotemporal variations in plantar pressure and the movement phases identified in the kinematic analysis, the entire motion sequence was divided into four distinct phases for detailed examination.

(1)Initial Phase

During the initial phase, both feet displayed concentrated plantar-pressure distribution in the medial forefoot region.

The left foot exhibited a rapid increase in plantar pressure from zero, with pressure primarily concentrated in the medial forefoot area, corresponding to a flexed posture of the left lower-limb joints.

Similarly, the right foot showed pressure concentration in the medial toe region, with the center of plantar pressure (COP) located near the toes, and the right lower limb also maintained a flexed position.

Overall, this phase reflected that both feet relied on the medial forefoot as the primary load-bearing area during initial contact to establish a stable support base.

(2)Double-Leg Support Gliding Phase

In the double-leg support gliding phase, the athlete transitioned from full left-foot contact to a stable dual-support sliding motion, during which the plantar-pressure distribution exhibited dynamic changes.

For the left foot, the pressure area expanded gradually from the toe region to the medial forefoot and heel regions. The heel contact area increased notably, and the COP shifted to a position slightly posterior to the midfoot. As sliding continued and the slideboard reached the end stopper, the left foot’s pressure distribution shifted markedly—from the medial forefoot to the lateral midfoot and heel regions. The COP further migrated toward the heel, indicating a redistribution of plantar loading after the cessation of gliding motion.

The right foot showed a somewhat different distribution pattern during this phase. Its pressure extended progressively from the toe region to the medial forefoot and heel, forming a three-zone load-bearing pattern. When the right foot made contact with the end stopper, the dual-leg sliding was interrupted, and the pressure region expanded toward the lateral midfoot, while the high-pressure zone in the medial forefoot diminished. The COP migrated to the central midfoot, whereas the medial toe region remained a high-pressure area, reflecting residual forward momentum at the braking phase.

(3)Single-Leg Support Phase

During the single-leg support phase, the sliding motion had completely stopped, and the athlete shifted from double-leg to single-leg weight bearing. The plantar-pressure distribution showed a distinct lateral load-bearing pattern.

For the left foot, pressure was mainly concentrated in the lateral midfoot region, with the COP located slightly laterally, indicating that inertial forces continued to influence the load-bearing pattern and that the lateral border of the foot served as the main supporting area. As the push-off movement began, the COP gradually migrated toward the central midfoot, although the lateral region remained the dominant load-bearing zone.

The right foot exhibited a similar pattern in this phase, with pressure concentrated in the lateral midfoot and the COP located near the center of the foot. During the initiation of the push-off motion, the lateral plantar region remained the primary load-bearing area.

Overall, both feet showed laterally shifted pressure centers and dominant lateral loading during single-leg support, suggesting that the early stage of push-off was still influenced by inertial effects and the foot’s lateral load-bearing characteristics.

(4)Push-Off Phase

In the push-off phase, the athlete entered the power generation, terminal, and completion stages of propulsion, during which plantar pressure exhibited a rapid rise to peak values followed by a sharp decline to near zero.

For the left foot, plantar pressure increased sharply to its maximum value (302.1 N) during the push-off peak, with pressure mainly distributed in the medial forefoot and heel regions. The COP was positioned slightly anterior to the midfoot, indicating that the main propulsive force was generated through the medial forefoot. As the push-off progressed, plantar pressure decreased rapidly; although the pressure remained concentrated in the medial forefoot, the area of high pressure became smaller, and the intensity decreased. The COP stayed in the anterior midfoot, reflecting attenuation of the propulsive force. By the end of push-off, plantar pressure dropped close to zero, with the remaining pressure concentrated at the toes and medial forefoot edge, marking the completion of support and propulsion by the left leg.

The right foot exhibited a similar pressure pattern throughout the push-off phase. At the peak of propulsion, the plantar pressure reached its maximum value (335.5 N), mainly concentrated in the forefoot, especially the medial forefoot region, with the COP located in this area, indicating focused power output. In the later stage, plantar pressure rapidly decreased; although the medial forefoot remained the main load-bearing region, both pressure magnitude and area decreased significantly, demonstrating a decline in propulsion. At the completion of the push-off, pressure approached zero and was mainly distributed along the toes and medial forefoot edge, with the COP remaining in this region. This pattern signified the end of the right-leg support and push-off action as the center of mass shifted back between both legs, initiating the next double-leg support gliding phase.

### 3.4. Results of the Cubic Polynomial Regression Model

Table 3 presented the results of the cubic polynomial regression analyses between lower-limb joint angles and ipsilateral plantar pressure during slideboard skating. The findings indicated that joint angles and plantar pressure exhibited varying degrees of nonlinear association. Substantial differences were observed in model fit across joints, with the highest goodness of fit identified in the model relating right hip flexion to right plantar pressure (R = 0.88, R^2^ = 0.77, adjusted R^2^ = 0.77). Conversely, the lowest fit occurred in the model relating left hip abduction to left plantar pressure (R = 0.52, R^2^ = 0.27, adjusted R^2^ = 0.27).

The contribution of each polynomial term differed across models. For instance, in the left hip flexion model, the first-order term exhibited a significant positive effect on plantar pressure (B = 1.09, 90% CI [0.10, 1.18], Beta = 1.09, t = 19.43, *p* < 0.001), the second-order term was also significant (B = 0.21, 90% CI [0.05, 0.36], Beta = 0.38, t = 2.21, *p* = 0.03), while the third-order term was non-significant (B = −0.01, 90% CI [−0.06, 0.04], Beta = 0.06, t = −0.37, *p* = 0.71). For the right hip, both first- and second-order terms were highly significant (*p* < 0.001), whereas the third-order term showed statistical significance but a negligible standardized effect size (B = 0.00, 90% CI [−0.05, 0.05], Beta = 0.01, t = 0.10, *p* < 0.001).

In the knee-flexion models, both the quadratic and cubic terms demonstrated strong negative associations with plantar pressure for the left knee (X^2^: B = −1.11, 90% CI [−1.22, −1.01], Beta = −1.96, t = −17.32, *p* < 0.001; X^3^: B = −0.48, 90% CI [−0.55, −0.41], Beta = −2.11, t = −11.08, *p* < 0.001), and a similar pattern was observed for the right knee. For ankle dorsiflexion, the first-order term was significantly positive for the left ankle (B = 1.11, 90% CI [0.94, 1.28], Beta = 1.11, t = 10.80, *p* < 0.001), the second-order term was non-significant, and the third-order term was significantly negative (B = −0.12, 90% CI [−0.22, −0.02], Beta = −0.42, t = −2.01, *p* = 0.05). For the right ankle, the first-order term was significantly positive, whereas both higher-order terms were significantly negative.

Model assumptions were evaluated using standardized residuals. Across all models, residual histograms displayed bell-shaped distributions centered near zero with gradually tapering tails, indicating no apparent skewness or extreme outliers and overall conformity to normality. Residual values ranged from −2.73 to 1.37, with a mean close to zero and a standard deviation of approximately one. Although some autocorrelation was present, the overall residual characteristics satisfied the fundamental assumptions of linear regression.

### 3.5. Symmetry Characteristics of Lower-Limb Kinematics and Plantar Pressure

The dynamic characteristics of plantar pressure and lower-limb joint angle trajectories, together with the results of the cubic polynomial regression analysis, revealed varying degrees of asymmetry between the left and right lower limbs in both kinematic and plantar loading characteristics. The mean plantar pressures of the left and right feet were 127.66 ± 94.84 N and 167.00 ± 125.68 N, respectively, indicating that the right foot generally bore a higher overall load during slideboard training. Similarly, the magnitudes of joint angle variations exhibited noticeable side-dependent differences.

To further characterize the symmetry of kinematic parameters and plantar loading during slideboard training, symmetry indices (SI%) were calculated for key joint angle metrics and plantar pressure variables. It should be emphasized that SI% was used to describe the relative magnitude of bilateral differences rather than to determine statistical significance between sides. Higher SI% values indicated greater bilateral asymmetry and lower symmetry. The results demonstrated substantial variability in SI% across different parameters (Figure 9). Among the plantar pressure variables, the SI% for peak pressure was 10.46%, whereas the SI% for mean pressure was considerably higher, reaching 26.70%. In contrast, the SI% for minimum pressure was 0%, indicating a high level of bilateral consistency in load magnitude at the end of the support phase. The SI% for the timing of peak pressure was 2.25%, suggesting a high degree of temporal synchrony between the two sides in the occurrence of maximal loading.

Among the joint angle parameters, the symmetry indices (SI%) for hip flexion were 0.19%, 10.44%, and 1.22% for peak angle, minimum angle, and mean angle, respectively. In contrast, the SI% values for range of motion and the timing of peak angle reached 23.70% and 13.70%, respectively, indicating a moderate degree of asymmetry in both amplitude regulation and temporal characteristics of hip flexion. By comparison, hip abduction exhibited substantially greater asymmetry, with SI% values of 88.52% and 37.95% for the minimum and mean angles, respectively, whereas the SI% for the peak abduction angle was relatively low (5.08%). These findings reflected pronounced bilateral differences in stance width and abduction–adduction control strategies between the lower limbs.

Knee flexion demonstrated the lowest SI% levels among all joint parameters, with SI% values for peak angle, minimum angle, mean angle, and time-related metrics ranging from 0.34% to 3.46%, indicating a high degree of bilateral symmetry during slideboard training.

For ankle dorsiflexion, the SI% values for peak angle and mean angle were 25.06% and 27.79%, respectively. The minimum angle was not expressed as a percentage because the left ankle exhibited plantarflexion (−5.04°), whereas the right ankle remained in dorsiflexion (3.89°). This structural difference in movement direction rendered the proportional SI% metric inapplicable; therefore, the asymmetry was described using the absolute angular difference (8.93°).

Overall, the distribution of SI% across joints indicated that different joints assumed distinct roles in symmetry regulation during slideboard training. Knee flexion exhibited the highest level of bilateral consistency, whereas hip abduction showed the most pronounced asymmetry. Ankle dorsiflexion demonstrated moderate to relatively high asymmetry, reflecting functional differentiation among joints in the control of sliding stability and push-off modulation.

## 4. Discussion

### 4.1. Initial Phase

In the initial phase, when the feet first made contact with the slideboard, the lower limbs exhibited a pronounced pre-loading posture characterized by substantial hip and knee flexion (hip flexion: left—108.7°; right—115.6°; knee flexion: left—95.8°; right—95.5°). This posture effectively lowered the body’s center of mass, thereby providing a stable foundation for subsequent movement. The degree of knee flexion was highly symmetrical between limbs, as indicated by the low SI% (peak angle = 6.94%; T_peak angle = 0.45%). This bilateral symmetry indicated that athletes had already established a well-balanced preparatory posture before meaningful load transfer occurred, which constituted a fundamental biomechanical prerequisite for postural stability during skating-related gliding tasks, as described in classical biomechanical analyses of skating movements [15]. In line with this general principle, Wu et al. demonstrated in their investigation of roller-skating biomechanics that bilateral stride symmetry contributed to the formation of a stable preparatory posture and balanced support during the early gliding phase [16], suggesting that symmetry during initial load acceptance represented a common biomechanical requirement across skating-related movement tasks rather than a discipline-specific feature.

Despite this overall symmetrical flexion strategy, hip abduction angles demonstrated marked asymmetry (left: −2.4°; right: 12.3°), indicating that lateral weight shifting had not yet taken place during this early phase. Because the slideboard did not generate forward momentum, mediolateral center-of-mass transfer was delayed; thus, the primary function of the lower limbs during initial contact was to maintain vertical stability rather than to perform the lateral balance adjustments that were critical during on-ice skating [17]. Prior research has shown that speed-skating performance relies heavily on precise center-of-mass control, as the extremely narrow blade contact surface amplifies even minor instability [18]. Consequently, achieving a stable center of mass was a prerequisite for effective propulsion. While slideboard training replicated certain technical components of skating, the lack of forward inertia during this initial phase limited the immediacy and magnitude of lateral weight shift. As a result, the early-phase movement strategy observed under slideboard conditions should be interpreted as a stability-oriented preparatory pattern, whose direct transferability to on-ice skating demands remained inherently constrained [19].

The plantar-pressure distribution further revealed differences between slideboard training and on-ice skating techniques. In the present study, the first appearance of plantar pressure occurred in the medial forefoot, showing a loading pattern that differed substantially from normal gait. Perry and Burnfield [20] reported that initial ground contact in typical gait usually occurred at the posterolateral heel, and Ma Guodong et al. [21] similarly observed in inline-skating tasks that athletes tended to establish initial support through the heel region. The medial-forefoot-dominant initial loading observed in this study reflected the unique mechanical demands of slideboard training: in the absence of forward momentum, athletes were required to rapidly absorb impact and establish support through the forefoot. Prior research has demonstrated that plantar-pressure distribution is closely associated with technical execution characteristics and can serve as a sensitive biomechanical indicator of movement effectiveness and control [22,23]. This forefoot-dominant loading strategy suggested not only a reorganization of lower-limb control specific to the slideboard environment but also highlighted a fundamental difference from the mechanical requirements of on-ice skating [24]. Although athletes were able to replicate the low-posture characteristics of ice skating through increased joint flexion, the altered plantar-pressure distribution indicated an inconsistency in mechanical execution [25]. Such discrepancies may have limited the transferability of slideboard training to actual on-ice performance and represented a key factor influencing the effectiveness of off-ice to on-ice technical transfer.

### 4.2. Double-Leg Support Gliding Phase

During the double-leg gliding phase, both the hip and knee joints showed a gradual decrease in flexion angles as the center of mass (COM) shifted toward the supporting limb. The change was most pronounced in the knee joint—decreasing from 95.8° to 73.5° on the left side and from 95.5° to 78.0° on the right—resulting in a noticeable elevation of body posture. From a biomechanical perspective, such an elevation in posture altered the orientation of force application during the support phase and may have reduced the proportion of force effectively directed along the intended sliding direction. Meanwhile, hip abduction angles increased continuously (left: −2.4° → 18.3°; right: 12.3° → 21.1°), indicating the onset of lateral shifting toward the supporting limb.

The increase in plantar pressure was relatively modest (left: +121.0 N; right: +117.2 N), suggesting that body weight had not yet been fully transferred to the right supporting leg. The plantar-pressure distribution was primarily concentrated in the medial toe region, medial forefoot, and heel, with minimal differences between sides.

Taken together, these findings indicated that athletes performed the lateral gliding movement with both legs sharing the load, forming a distinct triangular support structure. This observation was consistent with Li et al. [18], who noted that plantar-pressure patterns could effectively reflect the technical soundness and postural stability of skating movements. By integrating plantar pressure, joint kinematics, and spatial distribution characteristics, it was inferred that the COM remained positioned between the two lower limbs, with the bilateral triangular support structure bearing the majority of the gliding load (Figure 10). This represented a key mechanical feature of the double-leg gliding phase during slideboard training.

Existing research has shown that during slideboard training, the distance between the push-off and support legs tends to be greater and foot rotation angles are markedly smaller, which collectively restrict forward propulsion and explosive knee/ankle extension [26]. In the present study, the triangular support structure observed during the double-leg gliding phase was consistent with the typical movement characteristics reported for slideboard-based skating simulations. Due to the relatively high friction of the slideboard surface and the absence of forward inertia, athletes were unable to perform anticipatory adjustments of the center of mass prior to gliding—an ability that fundamentally distinguishes on-ice straight-line skating [25]. As a result, slideboard movement during this phase represented a predominantly passive balance strategy focused on maintaining longitudinal stability, while lateral motion was constrained by the bilateral load-sharing mechanism. However, such delayed lateral weight transfer and limited load-shifting capacity introduced constraints on movement coordination, thereby reducing the effectiveness with which slideboard training could transfer to actual on-ice skating performance. Krumm et al. indicated that the higher surface friction in slideboard imitation drills constrained lateral push-off forces, leading to delayed weight transfer and greater reliance on passive balance strategies, which in turn limited the transferability of technical skills to on-ice speed skating [1].

When the foot made contact with the terminal stopper on the slideboard, a distinct shift in plantar-pressure distribution was observed, characterized by a medial-to-lateral transfer of load. The center of pressure (COP) for both feet relocated toward the lateral midfoot region. This pattern indicated that once the sliding motion was abruptly halted by the stopper, the body’s center of mass was passively displaced laterally under the influence of residual inertia [27]. As a result, the subsequent single-leg support phase relied more on externally induced reactions than on active postural control. Zhang et al. [28] reported that although the gliding phase in speed skating does not directly generate propulsive force, small technical inefficiencies accumulated during this phase may still increase energy expenditure and reduce overall skating speed. Similarly, in slideboard training, the delayed center-of-mass adjustment and the reliance on passive balancing during the double-leg gliding phase may introduce comparable accumulative inefficiencies [29]. Li et al. [18] emphasized that maintaining a stable center of mass is essential for balance and directional control during on-ice skating. Because slideboard training lacked the low-friction environment and inherent inertia of actual ice skating, lateral weight transfer became constrained, making the passive balancing strategy observed at this stage less transferable to real skating performance.

### 4.3. Single-Leg Support Phase

Once entering the single-leg support phase, the athlete transitioned from bilateral to unilateral loading, with the supporting limb gradually bearing the entire body weight. Plantar pressure continued to rise and reached a distinct peak, while the center of pressure (COP) shifted toward the lateral midfoot [30]. This occurred because, after the sliding motion was halted by the terminal stopper on the slideboard, the body’s residual lateral inertia drove the center of mass toward the supporting leg, forcing the athlete to rely on the lateral foot region to maintain balance.

This inertia-induced shift also led to further increases in hip and knee flexion before these angles stabilized, whereas the ankle remained in a relatively high dorsiflexion position [31]. Consequently, the single-leg support posture became noticeably lower. Previous research has noted that increased knee flexion in speed skating reduces body height and aerodynamic drag, thereby enhancing gliding stability [24]. Consistent with this mechanism, the pronounced knee flexion and lowered posture observed during the single-leg support phase of slideboard training may facilitate the adoption of a similarly low, streamlined posture during on-ice straight-line gliding.

As plantar pressure increased, the hip abduction angle progressively decreased (left: 18.3° → 1.5°; right: 21.1° → 1.3°), indicating a clear medial shift in the center of mass from a bilateral stance toward the supporting limb. In the late single-leg support phase—specifically during the transition into the push-off action—the hip abduction angle reached its minimum (left: −3.0°, right: −1.6°) before beginning to rise again, while plantar pressure sharply declined and formed a distinct trough. Throughout this interval, the remaining lower-limb joints maintained relatively stable flexion angles, suggesting that lateral control of the center of mass was governed predominantly by variations in hip abduction. This pattern revealed a pronounced nonlinear coupling between hip abduction dynamics and plantar loading. Moreover, the hip abduction angle demonstrated a high symmetry index (SI%) for the mean angle (37.28%), and the regression models showed substantial bilateral differences (left R^2^ = 0.274; right R^2^ = 0.838). Taken together, these findings demonstrated pronounced asymmetry in hip abduction control and highlighted instability in lateral balance regulation and plantar support during the gliding phase of slideboard skating. Liu et al. demonstrated that bilateral asymmetries in hip angles during speed skating, particularly in the single-leg gliding phase, lead to notable instability in lateral center-of-mass control and reduced postural symmetry [32].

In speed skating, maintaining a stable center of mass is essential for balance and directional control, as the extremely narrow blade dramatically amplifies even minor postural instabilities, resulting in deviations in the skating trajectory [18]. Therefore, insufficient plantar support stability and lateral control of the center of mass during slideboard training may be transferred to on-ice skating, where such instability would be further amplified during single-leg gliding.

In the transition from single-leg support to the push-off phase, plantar pressure was primarily concentrated in the lateral midfoot and heel regions, with both the magnitude and contact area being greater in the lateral midfoot than in the heel [33]. This pattern indicated that athletes relied predominantly on the lateral midfoot to maintain stability and initiate the subsequent extension movement. However, such a pressure distribution may not correspond to a mechanically favorable push-off configuration and could indicate an alternative pressure-loading pattern associated with altered propulsion mechanics.

### 4.4. Push-Off Phase

The findings of this study indicated that the push-off phase during slideboard training was characterized by a typical power-generation pattern consisting of a low preparatory posture, lateral propulsion, and a rapid decline in plantar loading. The movement organization in this phase reflected a clear proprioceptive control strategy: the hip, knee, and ankle joints adopted a coordinated flexed posture at the onset of push-off, providing pre-stretch and energy-storage conditions, and subsequently released force sequentially through the kinetic chain in a hip–knee–ankle extension pattern. This pathway of force transmission closely resembled the kinetic-chain activation sequence observed in on-ice skating, indicating a high degree of technical similarity rather than complete mechanical equivalence. De Koning et al. reported that in speed skating, leg muscles exhibit a proximo-distal activation sequence during push-off, with coordinated hip, knee, and ankle extensions reflecting an efficient kinetic chain for force transmission and power output [34].

The hip joint played a dominant role during the push-off phase. Although its flexion–extension range was relatively limited, it contributed most substantially to increases in plantar pressure (R^2^: left = 0.62, right = 0.77; X_1_ coefficients: left B = 1.09, right B = 1.15), exhibiting a clear nonlinear response. This indicated that a hip-driven propulsion strategy remained the primary source of mechanical output during slideboard skating. Previous studies have shown that hip-extension capacity is strongly associated with skating speed [11]. Consistent with this perspective, the present findings highlighted the prominent role of the hip joint in lower-limb coordination and propulsion-related mechanics during off-ice slideboard training. Menegaldo et al. found that speed-skating athletes display adapted hip abduction and adduction torque–angle curves, with higher peak torques contributing dominantly to propulsion and mechanical work during the push-off cycle [35].

Additionally, the role of hip abduction differed markedly between sides. The right hip demonstrated efficient force transmission (R^2^ = 0.84; X_1_ B = 1.40, *p* < 0.001), whereas the left hip contributed minimally to propulsion (R^2^ = 0.27; X_1_ B = −0.50, *p* < 0.001), functioning primarily in center-of-mass control and postural stabilization. This asymmetrical regulation suggested that slideboard training places increased demands on lateral balance control and underscores the relevance of abduction–adduction mechanics in shaping propulsion-related movement coordination. Liu et al. observed significant bilateral asymmetries in hip angles during speed skating push-off phases, with differential contributions to force transmission and center-of-mass stabilization across sides [32].

The role of the knee joint also exhibited a distinct nonlinear relationship with plantar loading (R^2^: left = 0.57, right = 0.67; X_1_ coefficients: left B = 1.30, *p* < 0.001; right B = 0.73, *p* < 0.001). The substantial knee-flexion angles not only enhanced energy storage prior to push-off but also contributed to mediolateral control of the center of mass. The nonlinear nature of this relationship suggested that variations in knee flexion were associated with distinct force-transmission patterns during push-off, with different flexion magnitudes corresponding to altered coordination characteristics within the lower-limb kinetic chain. These findings were consistent with previous analyses of speed-skating propulsion mechanics [29], which similarly highlight the importance of controlled knee flexion for efficient lateral push-off.

The ankle joint also played a critical role during the late push-off phase. Ankle extension occurred only at the very end of propulsion, confirming that the majority of force generation throughout the movement was dominated by the hip and knee joints [36]. Although the ankle was not a primary contributor to overall power output, its terminal extension nonetheless served as an essential link in the force-transmission chain (R^2^: left = 0.50, right = 0.36; X_1_ coefficients: left B = 1.091, right B = 0.961). This contribution was most evident in the final release of force through the medial forefoot region. The observed loading pattern was highly consistent with the typical pressure-shift trajectory seen in on-ice skating, reflecting the effectiveness of slideboard training in reproducing short-range lateral propulsion and distal control mechanisms.

From the perspective of plantar-pressure distribution, the push-off phase exhibited a consistent “medial forefoot–dominant” loading pattern, with the center of pressure (COP) remaining concentrated in the medial region. This pattern reflected the combined effect of coordinated lower-limb joint extension and lateral–medial weight-shift control, which jointly drove the medial migration of plantar loading. Previous studies have suggested that plantar pressure serves as an important indicator of technical stability and movement efficiency [22]. In the present study, the COP trajectory consistently terminated in the medial forefoot, which aligned closely with the force-transmission pathway observed in on-ice skating propulsion. This finding indicated that despite the absence of low-friction conditions and continuous glide phases, the slideboard environment still preserved key biomechanical features of skating propulsion—including temporal sequencing, kinetic-chain activation patterns, and medial force-shift direction [37]. Therefore, the push-off mechanics exhibited during slideboard training demonstrated substantial technical transferability to on-ice performance.

## 5. Conclusions

This study developed a Kalman-based analytical framework that integrated inertial measurement unit (IMU)-based kinematic data with plantar-pressure information to characterize the coupling relationships between joint-angle trajectories and plantar loading during slideboard speed-skating training. The results demonstrated that a Kalman-based joint state-estimation approach was able to capture nonlinear coupling patterns between lower-limb joint angles and plantar-pressure variables across different movement phases, as reflected by consistently high coefficients of determination (R^2^ = 0.77–0.83, *p* < 0.01).

From an applied perspective, the present findings supported the feasibility of wearable sensor-based approaches for fine-grained analysis of lower-limb coordination in off-ice speed-skating training. The proposed framework provided quantitative descriptors of movement coordination and symmetry that may complement traditional visual assessments. However, the present analyses were intended to describe associative and coordination characteristics rather than to evaluate training outcomes or performance improvements.

Nevertheless, several limitations should be acknowledged. First, the relatively small sample size may limit the generalizability of the findings. Second, machine-learning-based feature extraction or predictive modeling approaches were not incorporated [38,39,40]. This study employed regression analysis to quantify associative relationships and model goodness of fit, rather than to develop predictive models or evaluate generalization performance. Future studies should further refine the data-fusion algorithms, recruit larger and more diverse participant cohorts, and develop portable real-time feedback systems capable of on-site data integration and visualization.

Overall, this work represented an exploratory investigation into the quantitative characterization of lower-limb coordination during slideboard speed-skating training and contributed methodological insight into the application of multimodal wearable-sensor analysis in sports biomechanics.

## Figures and Tables

**Figure 1 sensors-26-00272-f001:**
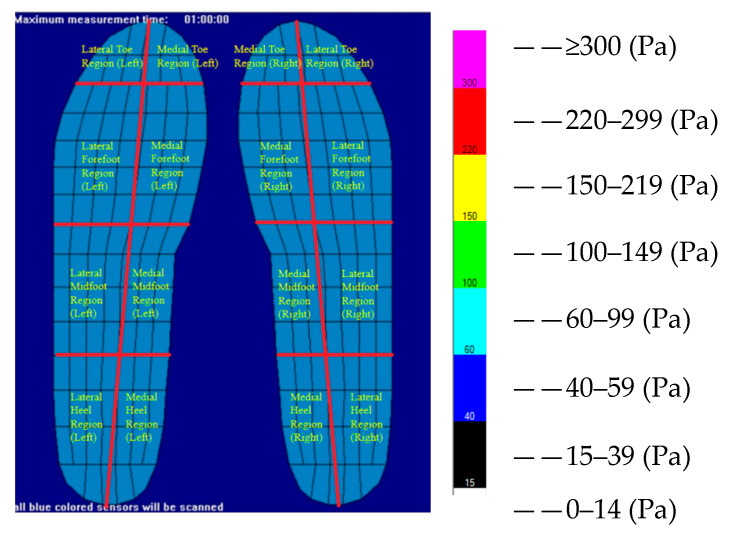
Plantar-pressure regional segmentation and pressure level classification.

**Figure 2 sensors-26-00272-f002:**
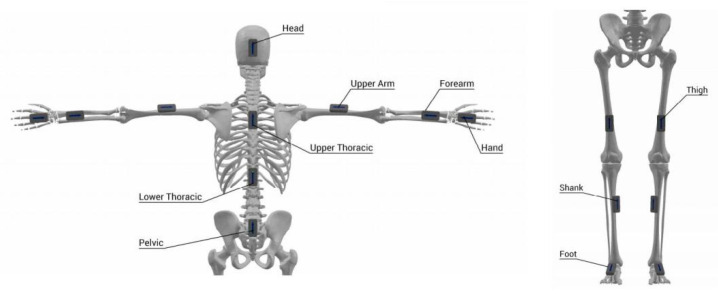
Schematic diagram of inertial sensor placement on the participant’s body.

**Figure 3 sensors-26-00272-f003:**
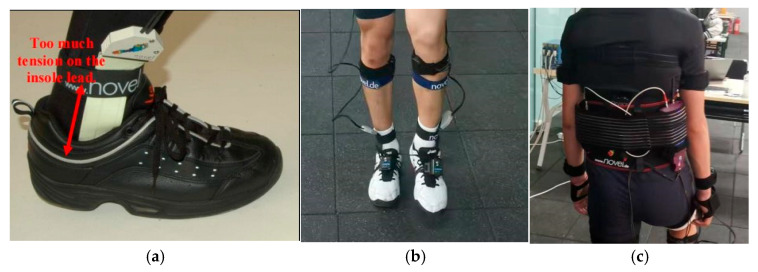
Pedar-X plantar-pressure measurement system and the experimental setup. (**a**) The plantar-pressure insole is placed inside the athletic shoe; (**b**) The connecting cables are secured to the lower limb using a specialized fixation strap; (**c**) The signal receiver and battery are secured to the back using a dedicated harness system.

**Figure 4 sensors-26-00272-f004:**
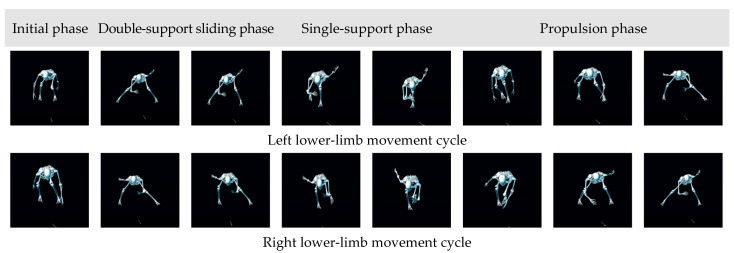
Definition of one gait cycle for the left and right lower limbs.

**Figure 5 sensors-26-00272-f005:**
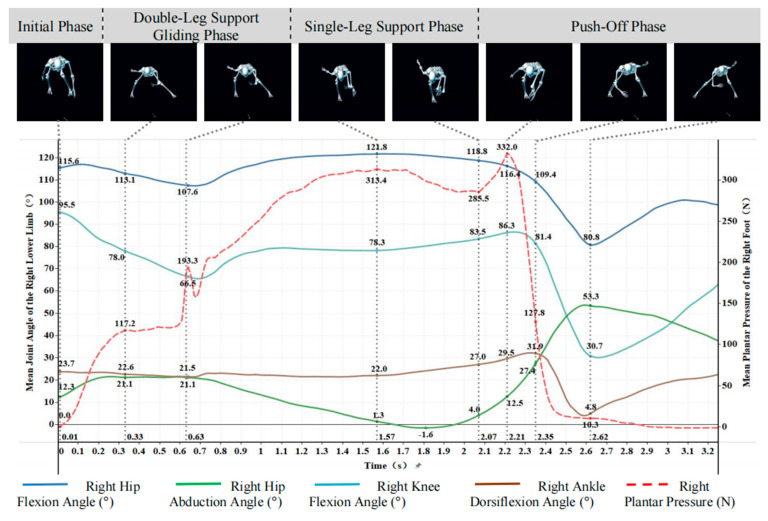
Curves of right lower-limb joint angles and right plantar pressure.

**Figure 6 sensors-26-00272-f006:**
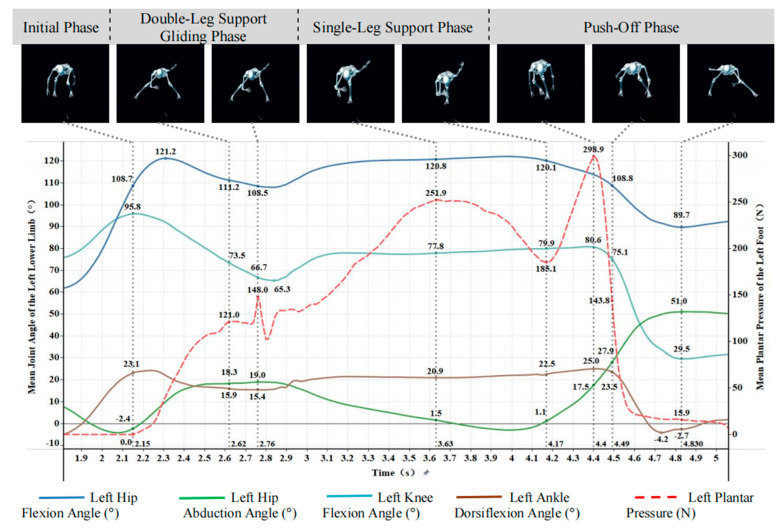
Curves of left lower-limb joint angles and left plantar pressure.

**Figure 7 sensors-26-00272-f007:**
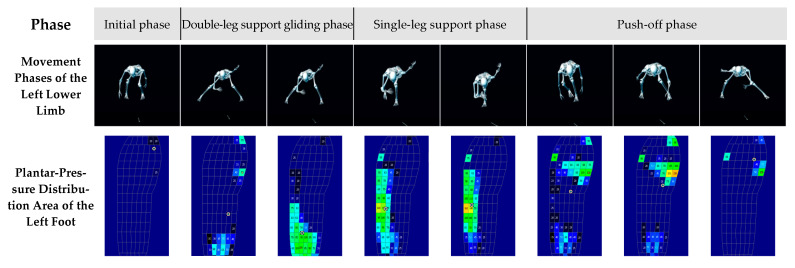
Dynamic characteristics of changes in the plantar-pressure distribution area of the left foot during slideboard training.

**Figure 8 sensors-26-00272-f008:**
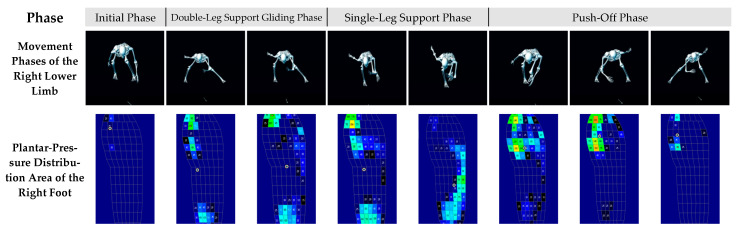
Dynamic characteristics of changes in the plantar-pressure distribution area of the right foot during slideboard training.

**Figure 9 sensors-26-00272-f009:**
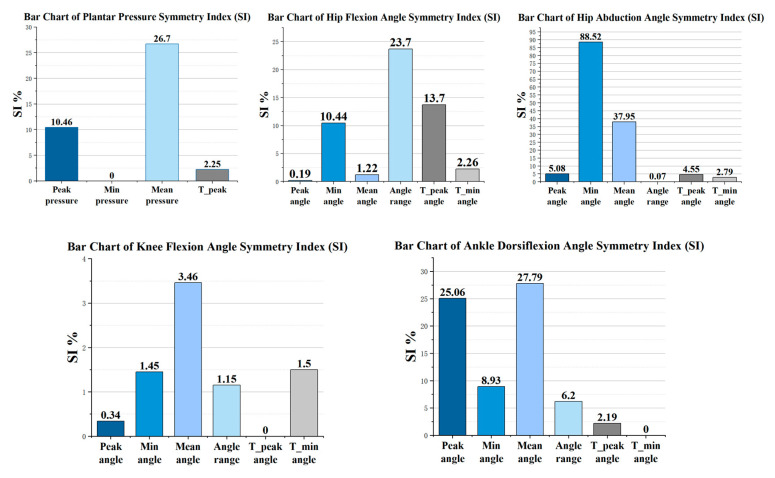
Summary of symmetry indices (SIs) for lower-limb kinematics and plantar pressure. Notes: Pressure peak: The maximum plantar-pressure value, representing the peak propulsive force generated during the push-off phase. Min pressure: The minimum plantar-pressure value, indicating the complete unloading or disappearance of foot-ground contact. Mean pressure: The average plantar pressure across the movement cycle. T_peak: The time point at which the peak plantar pressure occurs. Peak angle: The maximal joint flexion angle during the movement. Min angle: The minimal joint angle, reflecting the maximum joint extension reached. Mean angle: The average joint angle throughout the cycle. Angle range: The amplitude of joint-angle variation, representing the extent of flexion–extension movement. T_peak angle: The time point at which the maximal flexion angle occurs. T_min angle: The time point at which the minimal flexion angle (maximal extension) occurs.

**Figure 10 sensors-26-00272-f010:**
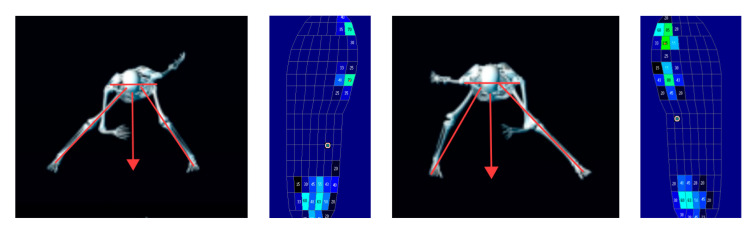
Center-of-mass position and plantar-pressure distribution during the double-leg gliding phase. Notes: The downward-pointing arrow denotes the location of the body’s center-of-mass projection.

**Table 1 sensors-26-00272-t001:** Basic information of the participants.

No.	Age (Years)	Height (cm)	Body Mass (kg)	Skeletal Muscle Mass (kg)	Body Fat (%)	Training Experience (Years)	Athletic Level
1	18	185	59.8	30.0	10.2	7	National Master Athlete
2	17	185	62.6	32.2	8.5	7	National Master Athlete
3	16	175	68.2	34.0	12.4	6	First-Level Athlete
4	16	185	71.1	38.1	6.1	5	First-Level Athlete
5	15	170	68.0	33.8	10.3	6	First-Level Athlete
6	17	173	56.8	30.3	6.5	6	National Master Athlete
7	16	175	67.5	33.1	10.3	6	First-Level Athlete
8	16	177	66.3	32.8	9.5	5	First-Level Athlete

Note: Values are presented as mean ± standard deviation (SD). All athletes were free of injury and participated in regular national-level training programs during the previous season.

**Table 2 sensors-26-00272-t002:** Characteristics of plantar pressure and lower-limb joint angles for the left and right sides.

Parameter	N_sample_	Min	Max	Range	Mean ± SD
Left plantar pressure (N)	326	0.00	298.94	298.94	127.66 ± 94.84
Right plantar pressure (N)	325	0.00	331.95	331.95	167.00 ± 125.68
Left hip flexion angle (°)	326	89.67	122.09	32.42	109.11 ± 14.94
Right hip flexion angle (°)	325	80.73	121.85	41.12	110.45 ± 11.53
Left hip abduction angle (°)	326	−4.24	51.01	55.25	14.38 ± 17.01
Right hip abduction angle (°)	325	−1.64	53.65	55.29	21.12 ± 17.57
Left knee flexion angle (°)	326	29.51	95.85	66.34	72.38 ± 17.53
Right knee flexion angle (°)	325	29.94	95.52	65.58	69.92 ± 17.16
Left ankle dorsiflexion angle (°)	326	−5.04	24.97	30.02	16.22 ± 8.75
Right ankle dorsiflexion angle (°)	325	3.89	32.09	28.20	21.46 ± 5.52

Note: Values are presented as mean ± standard deviation (SD). Negative values indicate movement in the opposite direction of the defined anatomical axis.

**Table 3 sensors-26-00272-t003:** Results of cubic polynomial regression analysis between lower-limb joint angles and plantar pressure during slideboard training.

Dependent Variable	Independent Variable (Order)	R	R^2^	Adj. R^2^	B	Beta	t	*p*	90% CI
Left plantar pressure	Left hip flexion X_1_	0.78	0.62	0.62	1.09	1.09	19.43	<0.001	0.10, 1.18
	Left hip flexion X_2_				0.21	0.38	2.21	0.03	0.05, 0.36
	Left hip flexion X_3_				−0.01	−0.06	−0.37	0.71	−0.06, 0.04
Right plantar pressure	Right hip flexion X_1_	0.88	0.77	0.77	1.15	1.15	22.97	<0.001	1.07, 1.24
	Right hip flexion X_2_				0.33	0.46	5.12	<0.001	0.22, 0.43
	Right hip flexion X_3_				0	0.01	0.1	<0.001	−0.05, 0.05
Left plantar pressure	Left hip abduction X_1_	0.52	0.27	0.27	−0.50	−0.50	−3.25	0.001	−0.75, −0.25
	Left hip abduction X_2_				−0.43	−0.61	−3.41	<0.001	−0.63, −0.22
	Left hip abduction X_3_				0.16	0.54	1.89	0.06	0.02, 0.36
Right plantar pressure	Right hip abduction X_1_	0.92	0.84	0.84	−1.40	−1.40	−19.65	<0.001	−1.52, −1.29
	Right hip abduction X_2_				0.03	0.03	0.83	<0.001	−0.03, 0.08
	Right hip abduction X_3_				0.24	0.53	6.39	<0.001	0.18, 0.31
Left plantar pressure	Left knee flexion X_1_	0.75	0.57	0.56	0.73	0.73	6.13	<0.001	0.53, 0.93
	Left knee flexion X_2_				−1.11	−1.96	−17.32	<0.001	−1.22, −1.01
	Left knee flexion X_3_				−0.48	−2.11	−11.08	<0.001	−0.55, −0.41
Right plantar pressure	Right knee flexion X_1_	0.82	0.67	0.67	1.3	1.3	17.67	<0.001	1.18, 1.42
	Right knee flexion X_2_				−0.73	−1.03	−12.74	<0.001	−0.82, −0.63
	Right knee flexion X_3_				−0.47	−1.58	−13.96	<0.001	−0.53, −0.42
Left plantar pressure	Left ankle dorsiflexion X_1_	0.71	0.50	0.50	1.11	1.11	10.8	<0.001	0.94, 1.28
	Left ankle dorsiflexion X_2_				0.03	0.05	0.28	0.781	−0.16, 0.22
	Left ankle dorsiflexion X_3_				−0.12	−0.42	−2.01	0.045	−0.22, −0.02
Right plantar pressure	Right ankle dorsiflexion X_1_	0.60	0.36	0.35	0.97	0.97	10.03	<0.001	0.81, 1.12
	Right ankle dorsiflexion X_2_				−0.18	−0.37	−4.54	<0.001	−0.23, −0.12
	Right ankle dorsiflexion X_3_				−0.13	−0.74	−5.67	<0.001	−0.16, −0.09

Notes: R represents the correlation coefficient; R^2^ denotes the coefficient of determination; adjusted R^2^ refers to the coefficient of determination adjusted for the number of predictors. B indicates the unstandardized regression coefficient, whereas Beta represents the standardized regression coefficient. t is the t-statistic associated with the coefficient, and *p* denotes the significance level; 90% CI lower and 90% CI upper indicate the lower and upper bounds of the 90% confidence interval for each regression coefficient, respectively.

## Data Availability

Data supporting the findings of this study are available from the corresponding author upon reasonable request.

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
