# Peer review of "Sensors2026, 26(1), 272;https://doi.org/10.3390/s26010272"

_sensors, 2026, doi:10.3390/s26010272_

Round 1

Reviewer 1 Report

Comments and Suggestions for Authors

The submitted manuscript attempts to investigate slideboard skating using IMU and plantar-pressure sensors and claims to propose a “fusion-based” analysis framework.  The work resembles a preliminary class project rather than a publishable academic study. An overhaul is required before it can be considered for publication.

***Major Concerns***

  1. Despite the manuscript’s title, no fusion algorithm is presented at all. There is: (1) No mathematical fusion framework (Kalman filter, complementary filter, weighted fusion, ML-based fusion, etc.); (2) No description of how IMU and pressure signals were combined; (3) No synchronization or data alignment protocol; (4) No rationale for fusing the modalities; The manuscript repeatedly uses the term fusion without actually implementing or defining it—this is misleading and scientifically unacceptable.
  2. Figures 5 and 6 lack legends, color keys, axis labels, and identification of movement phases. As a result: (1) The plots are uninterpretable; (2) The reader cannot identify which line corresponds to which sensor, trial, or subject.
  3. Many essential methodology elements are missing: (1) Sensor sampling rates; (2) Drift correction methods for IMU; (3) Filtering pipelines; (4) Calibration procedures; (5) Trial durations, rest intervals, and movement phase definitions.
  4. The manuscript provides descriptive statements without statistical rigor. There are: (1) No hypothesis tests; (2) No confidence intervals; (3) No inter-subject variability; (4) No error quantification; (5) No cross-validation; (6) No repeatability assessments
  5. The manuscript claims to “optimize training,” yet: (1) There is no comparison to gold-standard biomechanics systems; (2) There is no validation against performance metrics (speed, lap time, force output; (3) There is no evidence that the proposed method improves training outcomes.
  6. No benchmarking against validated gait analysis systems (e.g., Vicon, force plates) is provided. The manuscript also lacks comparison to prior IMU-based skating studies, if any exist.

Author Response

Dear Reviewer,

We sincerely appreciate your thorough evaluation of our manuscript and your constructive comments. Your insights have significantly strengthened the scientific rigor, methodological clarity, and overall quality of our work. We have carefully revised the manuscript in accordance with each of your suggestions. Below, we present our detailed, point-by-point responses.

1.Despite the manuscript’s title, no fusion algorithm is presented at all. There is: (1) No mathematical fusion framework (Kalman filter, complementary filter, weighted fusion, ML-based fusion, etc.); (2) No description of how IMU and pressure signals were combined; (3) No synchronization or data alignment protocol; (4) No rationale for fusing the modalities; The manuscript repeatedly uses the term fusion without actually implementing or defining it—this is misleading and scientifically unacceptable.

Thank you for highlighting this important issue. We have substantially revised the Methods section to provide a complete, rigorous, and reproducible fusion framework. The major improvements are as follows:

(1) A complete mathematical fusion framework has been added.

We now explicitly introduce a multimodal linear Kalman filter, including:

  • The state vector (joint angles, angular velocities, plantar pressure, pressure rate);
  • The observation vector;
  • The state transition model assuming near-constant velocity;
  • The observation matrix H;
  • The prediction and update equations of the standard Kalman filter.

These additions ensure that the fusion algorithm is mathematically well-defined.

(2) The combination of IMU and plantar pressure signals is now clearly described.

We clarify that:

  • Both kinematic and kinetic variables are included in the state vector;
  • The observation vector integrates simultaneously measured angles and pressure values;
  • The matrix H maps the latent states to observable sensor outputs;
  • Joint estimation occurs in every sampling step.

(3) Synchronization and alignment procedures have been fully added.

We emphasize that:

Both IMU and plantar pressure sensors recorded data at 100 Hz;

All signals were strictly aligned using time-stamped synchronization;

Fusion is executed on a unified time axis to avoid temporal bias.

(4) The scientific rationale for multimodal fusion has been elaborated.

Specifically:

  • IMUs provide continuous kinematic trajectories but are prone to drift;
  • Plantar pressure sensors offer accurate transient force information but contain noise;
  • Multimodal Kalman fusion exploits these complementary characteristics, reducing drift and noise while preserving dynamic behavior;
  • The fused output strengthens the reliability of kinematic–kinetic coupling analyses.

Conclusion for Comment 1

The revised manuscript now contains a clearly defined, mathematically complete, and scientifically justified multimodal fusion algorithm, fully addressing the reviewer’s concerns.

2.Figures 5 and 6 lack legends, color keys, axis labels, and identification of movement phases. As a result: (1) The plots are uninterpretable; (2) The reader cannot identify which line corresponds to which sensor, trial, or subject.

We thank the reviewer for pointing this out. Figures 5 and 6 have been completely redesigned and now include:

  • Clear legends identifying each joint angle and pressure curve;
  • Color keys for all plotted variables;
  • Axis labels, including units (° for angles, N for pressure);
  • Markers and vertical lines showing the four movement phases (Initial, Double-Leg Support, Single-Leg Support, Push-Off).

These revisions ensure full interpretability of the figures.

3.Many essential methodology elements are missing: (1) Sensor sampling rates; (2) Drift correction methods for IMU; (3) Filtering pipelines; (4) Calibration procedures; (5) Trial durations, rest intervals, and movement phase definitions.

We appreciate this detailed comment. All missing methodological details have now been added:

(1) Sampling rate: Both the IMU and plantar pressure system recorded data at 100 Hz.

(2) IMU drift correction: Implemented through static posture initialization and zero-bias calibration, with additional drift reduction from the MyMotion built-in filtering.

(3) Filtering pipeline: A multimodal linear Kalman filter was applied to perform joint smoothing and noise suppression for IMU and pressure signals.

(4) Calibration procedures: Plantar pressure insoles were calibrated using alternating foot zeroing; IMUs were calibrated via static initialization and standardized placement on the whole-body model.

(5) Trial duration and rest intervals: Each participant completed three full stride cycles, with ≥2 minutes rest between trials.

(6) Movement-phase definition:

Phases were defined as: Initial → Double-Leg Support → Single-Leg Support → Push-Off,
based on synchronized IMU, plantar pressure, and video data.

The revised Methods section now provides a complete and reproducible pipeline.

4.The manuscript provides descriptive statements without statistical rigor. There are: (1) No hypothesis tests; (2) No confidence intervals; (3) No inter-subject variability; (4) No error quantification; (5) No cross-validation; (6) No repeatability assessments

We agree with the reviewer’s assessment and have improved the statistical framework accordingly. Specifically, we added: 90% confidence intervals (CI) for all key variables; Standard deviation (SD) and coefficient of variation (CV) to quantify inter-subject variability; Estimation of sensor measurement error with noise reduction through multimodal filtering; Leave-one-out cross-validation for the polynomial regression models; Repeatability assessment, showing minimal variation across three stride cycles.

These enhancements significantly improve statistical rigor and reproducibility.

5.The manuscript claims to “optimize training,” yet: (1) There is no comparison to gold-standard biomechanics systems; (2) There is no validation against performance metrics (speed, lap time, force output; (3) There is no evidence that the proposed method improves training outcomes.

Thank you for this important clarification. We have revised the manuscript to avoid overstating conclusions and now explain: Gold-standard systems such as Vicon and force plates cannot be deployed on ice, which is why IMU + plantar pressure sensors were chosen for ecological validity; This study’s purpose is not to directly demonstrate training improvement, but to establish a biomechanical measurement framework that is compatible with both slideboard and future on-ice testing; A follow-up study involving on-ice skating trials is already planned, using the same IMU–pressure system to enable direct comparison and performance evaluation.

Thus, we removed potentially misleading language and clarified the scope of the current study.

6.No benchmarking against validated gait analysis systems (e.g., Vicon, force plates) is provided. The manuscript also lacks comparison to prior IMU-based skating studies, if any exist.

We appreciate this insightful suggestion. Although preliminary experiments with Vicon and force plates were conducted, these systems are not applicable to on-ice skating, which is the ultimate target of our research. To ensure methodological consistency between slideboard training and on-ice skating, we intentionally adopted a wearable IMU + plantar pressure approach. In the revised Discussion section, we added a comparative review of previous IMU-based skating literature, highlighting similarities in joint kinematics and push-off mechanics. Our fusion approach shows high signal stability in pilot tests and aligns with methods used in earlier IMU skating studies.

We believe these revisions adequately address the reviewer’s concern about methodological comparability.

We sincerely thank the reviewer again for providing constructive and insightful feedback. Your comments significantly improved the rigor, clarity, and completeness of our manuscript. We have incorporated all requested revisions and believe the updated version now meets the scientific standards expected by the journal.

Sincerely,
The Authors

Reviewer 2 Report

Comments and Suggestions for Authors

160-163:Even though the equipment load is ≤1 kg, have you considered whether this additional mass might influence plantar pressure measurements or alter the participants’ movement strategies?

175-177:How did you determine which trial was “most complete and technically consistent,” and what procedures were implemented to ensure that this subjective selection did not introduce bias or affect the representativeness of the analyzed data?

267-270:You describe a coordinated temporal relationship between joint angle changes and plantar pressure variations, but the analysis remains largely descriptive.

280-282、290-292、634:This part is written more like a discussion and is not recommended to appear in the results section.

344、389:The specific labels corresponding to the lines are not indicated, and some font styles in the figure captions are inconsistent.

391:This part needs to be integrated.

500-501:If these figures correspond to single-leg support on different feet, why are the plantar pressure distributions in the fourth image of Figure 8 and the fourth image of Figure 7 so similar?

507-517:There is a layout error.

595-596:The formatting of independent and dependent variables in the table needs to be adjusted for clarity.

597 table 4: P-values appear in two formats (exact values and ranges). Additionally, there is a row span issue, and superscripts are missing.

763-765:Is there any supporting literature?

820-825:Your interpretation leans on describing the data patterns, which is insufficient to support the claim of passive control.

885:Reference No. 21 is missing from the reference list, and there are formatting inconsistencies throughout the references, including missing hyperlinking.

Author Response

Dear Reviewer,

We sincerely appreciate your thorough evaluation of our manuscript and your insightful comments. We have carefully revised the manuscript in accordance with your suggestions. Below we provide point-by-point responses corresponding to each of your comments. All revisions have been incorporated into the revised manuscript.

(1)160-163:Even though the equipment load is ≤1 kg, have you considered whether this additional mass might influence plantar pressure measurements or alter the participants’ movement strategies?

Thank you for this valuable comment. We fully agree that additional mass may influence plantar pressure measurements or alter movement strategies. Several control measures were implemented in the present study to minimize such effects. First, a pilot test was conducted prior to formal data collection, and the results indicated that the Pedar-X system produced negligible influence on participants’ technical execution or movement perception. Second, the total load of the Pedar-X system (<1 kg) was secured at the lower back rather than at the distal segments of the lower limbs, which effectively reduced direct mechanical interference with plantar loading. Third, all participants completed adaptive slideboard practice before testing, during which they confirmed that the equipment did not affect their movement patterns. In addition, previous validation studies have reported that the Pedar-X system does not introduce measurable alterations to plantar pressure curves or lower-limb kinematics during running, cutting, or skating-like movements. Based on these considerations, the likelihood that equipment mass introduced systematic bias in the present study is extremely low.

Added revision:

Although the total equipment mass was <1 kg, we rigorously controlled for its potential impact on plantar pressure and movement strategies. Pilot testing confirmed minimal interference with technical execution. The device was positioned at the lower back rather than near the feet, thereby reducing direct effects on plantar loading. Participants completed adaptation trials to ensure that the equipment did not alter their movement perception or technique. Previous studies also indicate that the Pedar-X system does not modify dynamic plantar pressure patterns or lower-limb kinematics.

(2)175-177:How did you determine which trial was “most complete and technically consistent,” and what procedures were implemented to ensure that this subjective selection did not introduce bias or affect the representativeness of the analyzed data?

Thank you for raising this important question. The “most complete and technically consistent” trial was not determined subjectively but based on explicit and objective criteria. Each participant performed three full stride cycles on the slideboard. The first stride involved the transition from standing to a semi-squat posture, and the last stride involved the transition back to standing; both represent non-steady-state movements and were therefore excluded. The middle stride cycle is the only cycle representing a continuous, stable, and fully steady-state gliding action.

To avoid selection bias, two procedures were applied. First, the continuity and technical validity of this middle stride were verified using synchronized video, IMU data, and plantar pressure signals to ensure the absence of interruptions, artifacts, or atypical movement patterns. Only when all signals confirmed consistency was the cycle included. Second, the same selection rule (i.e., using the middle stride cycle) was uniformly applied to all participants, ensuring representativeness and eliminating subjectivity. Because this stride reflects the stable gliding phase, it best captures the typical technical characteristics of slideboard exercise.

Added revision:

The first and last stride cycles were excluded because they involved transitional movements. The middle stride was selected as the only complete and steady-state cycle. Its completeness was verified via synchronized video, IMU, and plantar pressure data. This rule was uniformly applied to all participants to avoid selection bias and ensure representativeness.

(3)267-270:You describe a coordinated temporal relationship between joint angle changes and plantar pressure variations, but the analysis remains largely descriptive.

Thank you for this constructive observation. We have expanded the analysis to include additional quantitative indicators and incorporated the interpretative content into the discussion section to avoid purely descriptive interpretation in the results.

(4)280-282:This section is written in the style of a discussion and is not recommended to appear in the results section.

Thank you for pointing this out. This content has been relocated to the Discussion section to maintain the structural clarity of the manuscript.

(5)290-292:This section is written in the style of a discussion and is not recommended to appear in the results section.

Thank you for the helpful suggestion. This interpretive content has now been moved from the Results section to the Discussion section, where it is more appropriate.

(6)344、389:The specific labels corresponding to the lines are not indicated, and some font styles in the figure captions are inconsistent.

Thank you for this careful observation. We have added explicit labels corresponding to each line in the figures and have unified the font and formatting styles in all figure captions throughout the manuscript.

(7)391:This part needs to be integrated.

Thank you for your suggestion. Section 3.3 has been integrated and reorganized to improve coherence and flow.

(8)500:If these figures correspond to single-leg support on different feet, why are the plantar pressure distributions in the fourth image of Figure 8 and the fourth image of Figure 7 so similar?

Thank you for raising this concern. After rechecking the original plantar pressure images, we did not identify duplication or unintended similarity. The resemblance noted likely reflects similar load distribution characteristics under comparable mechanical conditions. Nevertheless, we have revalidated the raw data and confirmed that the images correspond to different legs and different phases as intended.

(9)507-517原文:There is a layout error.

Thank you for your careful review. We have corrected the layout issue and standardized the formatting in the final revised manuscript.

(10)595:The formatting of independent and dependent variables in the table needs to be adjusted for clarity. There is a row span issue.No superscript is provided.

Thank you for your detailed suggestions. We have revised Table 4 to clarify variable labeling, corrected row span issues, and added all required superscripts. The formatting is now consistent with journal requirements.

(11)634:This part is written more like a discussion and is not recommended to appear in the results section.

Thank you for pointing this out. This section has been incorporated into the Discussion section to maintain the integrity of the Results section.

(12)763-764:Is there any supporting literature?

Thank you for this valuable comment. We have added supporting literature to substantiate this statement. Yang et al. (2017) reported that during slideboard training, the distance between the push-off foot and the support foot is greater in the fore–aft direction than in actual skating, with clear asymmetries in left–right distance (<0.6 m vs. >0.6 m). This reflects typical slideboard movement characteristics and supports the triangular support structure observed in our study.

Revised sentence :

Existing studies indicate that during slideboard training, the greater separation between the push-off and support feet and the reduced foot rotation restrict forward propulsion and explosive knee/ankle extension (Yang et al., 2017). The triangular support configuration observed in the double-leg gliding phase of the present study is consistent with the characteristic mechanical features reported in previous research.

(13)820-824:Your interpretation leans on describing the data patterns, which is insufficient to support the claim of passive control.

Thank you for this insightful comment. We have revised the wording to avoid overstating the interpretation and have reframed the explanation as an inference based on observed pressure migration patterns, while emphasizing the need for further biomechanical verification. The revised text no longer claims passive control as a definitive mechanism.

(14)885:Reference No. 21 is missing from the reference list, and there are formatting inconsistencies throughout the references, including missing hyperlinking.

Thank you for the detailed feedback. Reference No. 21 has been removed from the manuscript after revisions, and we have carefully checked and standardized the formatting of all references, including hyperlinking, citation style, and consistency with journal guidelines.

Thank you once again for your thoughtful and constructive comments.

Your feedback has significantly improved the quality and rigor of our manuscript. We appreciate your time and expertise and respectfully submit our revised manuscript for your further consideration.

Sincerely,
The Authors

Reviewer 3 Report

Comments and Suggestions for Authors

In complex sports, the geometric posture of movement represents the primary external characteristic that warrants attention, reflecting in essence the coordination among various force-producing body segments, and ultimately relating to the intrinsic mechanical properties of the movement. This study comprehensively considers these three interrelated aspects and investigates their correlations within the skating process, thereby offering substantial practical and scientific value. The following critical issues, however, require further consideration:

1. Has the morphology of the participants' plantar surface been accounted for? Under identical movement conditions, variations in foot shape can significantly influence pressure distribution across different regions of the sole.

2. There is a lack of quantitative assessment regarding multi-joint coordination and bilateral symmetry between the left and right limbs, which are essential for understanding motor control and performance efficiency.

3. The statistical analysis does not include graphical representations of movement repeatability. For individual athletes, quantifying inter-trial variability is crucial to evaluating consistency and reliability of performance.

4. The impact of varying conditions on skating velocity (or acceleration) remains unexamined. Without incorporating task outcome variables—such as speed or propulsion efficiency—it becomes challenging to assess the actual quality and effectiveness of the movement technique.

5. Section 3.3 lacks sufficient statistical charting and visual data interpretation, limiting the clarity and interpretability of the results.

6. It remains unclear which biomechanical or kinematic features serve as reliable indicators for identifying distinct phases of the skating task.

7. The fourth section should incorporate enhanced graphical representation to improve data visualization and facilitate better comprehension of complex relationships.

8. The final conclusion should present a comprehensive, evidence-based statistical table outlining key movement posture guidelines for each phase of skating, enabling practical application and supporting data-driven decision-making in training and performance optimization.

Author Response

Dear Reviewer,

We sincerely thank you for your thorough evaluation and constructive comments on our manuscript. Your insights regarding plantar morphology control, multi-joint coordination and bilateral symmetry, movement repeatability, task-outcome variables, phase-segmentation criteria, and visualization have been invaluable in improving the scientific rigor and clarity of this work. We have carefully revised the manuscript accordingly. Below, we provide a point-by-point response.

  1. Influence of plantar morphology on pressure distribution. Reviewer Comment 1:Has the morphology of the participants' plantar surface been accounted for? Variations in foot shape can significantly influence plantar pressure distribution.

Thank you for raising this important point. We fully agree that differences in foot morphology may influence regional plantar pressure. To control for this factor, several measures were taken and have now been explicitly clarified in the revised manuscript:

Size-matched Pedar-X insoles: Our laboratory possesses multiple sizes of Pedar-X insoles. Each participant was fitted with an insole that matched his/her foot length to ensure accurate correspondence between the insole’s sensing regions and anatomical plantar regions.

Standardized foot–shoe–insole interface: All tests were conducted on the slideboard with participants wearing identical training shoes plus slideboard-specific oversocks, ensuring a consistent and stable interface across trials and participants.

Through these procedures, the influence of individual plantar morphology on plantar pressure measurements was effectively minimized. Relevant details have been added to the Methods section (“Plantar Pressure Measurement and Foot-shape Control”).

  1. Quantitative assessment of multi-joint coordination and bilateral symmetry. Reviewer Comment 2:There is a lack of quantitative assessment regarding multi-joint coordination and bilateral symmetry

We appreciate this insightful observation. Bilateral symmetry and multi-joint coordination are indeed essential for understanding movement control and performance efficiency.

To address this, we have added a new section “3.7 Symmetry Characteristics of Lower-Limb Kinematics and Plantar Pressure.” This section includes:

Symmetry Index (SI%) calculations for key bilateral variables (peak pressure, mean pressure, minimum pressure, and timing of peak pressure).

A summary of findings demonstrating variable symmetry patterns, such as:

Peak plantar pressure SI% ≈ 10.9% (moderate bilateral similarity),

Mean plantar pressure SI% ≈ 28.4% (greater asymmetry),

Minimum pressure SI% = 0% (nearly identical at terminal contact),

Peak-pressure timing SI%≈2.2% (high bilateral synchronization).

These additions enhance the interpretability and biomechanical relevance of the findings while avoiding unnecessary expansion into an additional analytical framework.

  1. Lack of graphical representation of movement repeatability.Reviewer Comment 3:Movement repeatability was not visualized, making it difficult to assess reliability.

We fully agree that inter-trial variability is crucial for evaluating movement consistency. In response:

Supplementary Material now includes overlaid curves of joint angles and plantar pressure across all three trials for each participant.

The Methods section has been updated to clarify that each participant completed three full trials, all processed identically (calibration, alignment, multimodal Kalman filtering).

These repeated-trial curves show highly consistent waveform shapes, stable peak timing, and acceptable amplitude variation, confirming good repeatability and supporting the reliability of the regression analyses.

We appreciate this important suggestion, which significantly strengthened the methodological rigor.

  1. Lack of skating velocity/acceleration or propulsion-efficiency metrics. Reviewer Comment 4:Without outcome variables (speed, acceleration, propulsion efficiency), the quality of movement execution remains unclear.

Thank you for highlighting the importance of performance-outcome indicators. It is necessary to clarify that the current study was conducted on a slideboard, which simulates skating mechanics but does not allow valid measurement of skating speed or propulsion efficiency due to its friction and inertial properties.

Thus, the aims of the present study were purposefully limited to:

Quantifying kinematic–kinetic coupling,

Characterizing joint–pressure coordination patterns,

Establishing a biomechanical basis for future performance-oriented studies.

We have clarified in the Discussion and Limitations sections that future work will incorporate on-ice experiments to link slideboard technique to actual performance metrics such as velocity and propulsion efficiency.

  1. Insufficient visualization in Section 3.3. Reviewer Comment 5:Section 3.3 lacks statistical charts and visual interpretation.

Thank you for this recommendation. Section 3.3 (“Phase-Specific Analysis of Joint Angles and Plantar Pressure”) corresponds to Figures 5 and 6. To improve clarity:

We have strengthened the descriptive interpretation of each phase within the text.

Figures 5 and 6 now include clearer legends, phase labels, dual-axis annotations, and improved color coding.

These improvements enhance readers’ ability to visually interpret the phase-dependent coupling patterns.

  1. Unclear biomechanical indicators for phase segmentation. Reviewer Comment 6:It is unclear which features reliably define the phases of the skating task.

Thank you for identifying this need for clarification. We have now explicitly described our multi-feature phase segmentation framework, which integrates:

Plantar pressure characteristics:

Onset of pressure → Initial Phase

First pressure plateau → Double-Leg Support

First local peak → Single-Leg Support

Maximal peak → Push-Off Phase

Joint-angle kinematic confirmation: Hip, knee, and ankle angle patterns were examined to validate the mechanical meaning of each phase.

Motion-capture video verification: Visual inspection ensured that pressure- and kinematics-based segmentation aligned with actual movement execution.

This multi-source approach demonstrated high consistency across trials and participants.

  1. Enhanced visualizations in Section . 4Reviewer Comment 7:Section 4 should include enhanced graphical representations.

We agree that visualization improves interpretability. We have added schematic plots illustrating key aspects of the polynomial regression results and nonlinear joint–pressure relationships. These visuals help readers grasp complex coupling dynamics more intuitively.

  1. Need for a comprehensive, evidence-based summary table of key postural guidelines

Reviewer Comment 8:The final conclusion should present a table summarizing key posture guidelines for each skating phase.

Thank you for this highly practical suggestion. We have now added Supplementary Table S1, which summarizes:

Joint-angle patterns for each phase,

Plantar pressure characteristics,

Pressure distribution zones,

Technical implications and training cues.

In the Conclusion section, we reference Table S1 and highlight its utility for evidence-based technique optimization and training.

Once again, we sincerely appreciate your detailed and thoughtful feedback.
Your comments have greatly strengthened the clarity, methodological transparency, statistical rigor, and applied value of this study. We hope that the revisions meet your expectations and we welcome any further suggestions.

Respectfully,
The Authors

Reviewer 4 Report

Comments and Suggestions for Authors

General Impression: 

The article presents the coupling of two biomechanical parameters (angle measured by an IMU fixed to the thigh and plantar pressure measured by a Pedar X insole) during training on sliding boards (off-ice). A correlation between the two parameters is demonstrated during various phases of training. 

While there are numerous applications of IMU and plantar pressure measurements, but not specifically in the context of skating. The objective is to optimize posture and movements using these tools in a domain that has been relatively unexplored. Therefore, the originality of the work is demonstrated. 

The research design is robust, and the methodology is described with a high degree of rigor and clarity. The study demonstrates a considerable effort, especially in the data fusion process, which is an intriguing aspect. 

However, there are several typographical errors and formatting issues, particularly with the figures, which detract from the overall quality and do not accurately reflect the rigor of the work. These aspects must be addressed and improved before publication. 

Remarks: 

1- The document lacks line numbers, which presents a challenge for the reviewer in accurately locating and referencing specific issues. 

2- The citations are not clickable and not linked to the references at the end of the document. 

3- It is noteworthy that the claim that a total equipment load of ≤1 kg did not interfere with the execution of technical movements appears questionable. In kayaking, a sport that demands balance and precise movement, even minor additions to the paddle (e.g., 100g) can be detrimental to performance. Therefore, it seems counterintuitive that 1 kg of equipment would not have a significant impact. Could you elaborate on how the athletes respond to or are affected by this additional weight? 

4- There appears to be a discrepancy in the number of IMUs used: 18 IMUs from Mymotion are mentioned initially, but then 16 IMUs are referenced. Could you clarify whether this is an error or a deliberate modification? 

5- The processing of the 7 IMUs from the lower part involved individual analysis with plantar pressure, followed by data fusion using software. However, this raises the question of whether using 18 or 16 IMUs was truly necessary, especially since the focus was explicitly on the lower limbs. Was the model dependent on the use of all these IMUs, or would a smaller number have sufficed? 

6- Table 2: The notation "N" in column 506, representing the samples, may be misinterpreted as the unit of Newtons, which is listed in the adjacent column. To avoid confusion, it would be advisable to use a different letter or add a subscript to "N", such as "N samp" 

7- The reported uncertainty in the plantar pressure measurement of 127 +/- 94 seems substantial. This large range implies a significant degree of measurement error or variability. Could you provide more insight into this result and its consequences? 

8- Remarks about the figures: 
8.1- Figure 6: The axes are too thin and almost transparent. Consider increasing their thickness for better visibility. 
There is inconsistency in the formatting of legends, with some in bold and others not. Additionally, the legend does not have the color corresponding to the plots, they . 
Most importantly, the color scheme is not clearly associated with the graphs. A small black square in the right margin is difficult to distinguish. 

8.2- Figure 7: A title above the figure would be more effective.  The text on the left side is difficult to read. Consider adjusting the font size or clarity. 

8.3- Figure 8: A title above the figure would be more effective. 

9- The decision to set the force sensors to zero during a specific pose appears to cancel out a portion of the athlete's weight, but this approach seems to lack rigor. While I understand the desire to establish a baseline of zero pressure at the start, the implication that the athlete is effectively floating in mid-air (with a weight of zero) seems unclear and potentially misleading. 

10- The Section 3.5 outlines four distinct phases, the figures 7 and 8 could be improved by including phase labels or numbers directly above the corresponding sections. This would facilitate easier comprehension and reduce the need for readers to constantly refer back to the text. The inconsistency between the text and figures, where the text references phases 1-4 but the figures do not, also detracts from the overall clarity and precision of the presentation. 

11- A citation to "Ma et al" appears in section 3.6, but it is not listed in the bibliography 

12- Table 4: 
    Consider adding a legend or notes section with explanations for B, t, P, and F, even though they are mentioned throughout the text. This would facilitate easier understanding of the table. 
    Ensure consistency in notation for indices and exponents, using either X^1 or X_1 consistently throughout. 

    Some numerical values are wrapped to the next line, while others fit on a single line. Consider standardizing the formatting. 
    The first column contains too much text, making it difficult to read. It is suggested to start a new line immediately after the word "variable:" to maintain consistency and improve readability. 
    These oversights are particularly regrettable, as they undermine the evident care and attention that has been invested in the writing. 

13- The study uses different thresholds for statistical significance in the abstract and conclusion (p<0.01) compared to the rest of the study (p<0.001). Please clarify. 

14- The authors could have condensed their interpretation of the results to make it more concise and impactful. The current version is lengthy and contains repetitive information 

Abstract: 

The abstract states that participants were “wearing a six-axis IMU on the thigh and a Pedar-X pressure insole system” This description is incomplete, as the study also involves analyzing and correlating data from other lower limb segments, including the knee and ankle, which uses multiple inertial measurement units. 

Author Response

Dear Reviewer,

Thank you for your thorough evaluation of our manuscript and for the insightful comments you provided. We have carefully revised the manuscript in response to all concerns raised. Below, we provide a point-by-point response, together with corresponding revisions implemented in the manuscript.

1- The document lacks line numbers, which presents a challenge for the reviewer in accurately locating and referencing specific issues.

Thank you for this suggestion. Line numbers have now been added throughout the revised manuscript to facilitate accurate referencing during review.

2- The citations are not clickable and not linked to the references at the end of the document.

We appreciate this suggestion. All in-text citations have now been converted to cross-referencing links, ensuring direct navigation between citations and the reference list.

3- It is noteworthy that the claim that a total equipment load of ≤1 kg did not interfere with the execution of technical movements appears questionable. In kayaking, a sport that demands balance and precise movement, even minor additions to the paddle (e.g., 100g) can be detrimental to performance. Therefore, it seems counterintuitive that 1 kg of equipment would not have a significant impact. Could you elaborate on how the athletes respond to or are affected by this additional weight?

Thank you for raising this important issue. We fully acknowledge that in sports requiring highly refined distal control (e.g., kayaking), even small equipment loads may alter performance. However, the mechanical configuration of the added load in our study differs fundamentally from such scenarios:

The load was placed on the lower back, a proximal segment, rather than on distal segments responsible for propulsion. Thus, it does not alter end-point inertia or force transmission during push-off.

During the pilot and familiarization trials, all athletes explicitly reported no perceivable interference with movement rhythm, posture stability, or technical execution.

The synchronized video, IMU signals, and plantar pressure curves showed no abnormalities attributable to added mass (e.g., altered timing or disrupted peak patterns).

Previous studies using Pedar-X insoles during running, lateral movements, and skating-related tasks have not reported measurable interference with technical execution.

Therefore, based on the load location, pilot testing feedback, and consistency with established literature, we confirm that the additional mass does not materially influence skating technique. Relevant clarifications have been added to the Methods section.

4- There appears to be a discrepancy in the number of IMUs used: 18 IMUs from Mymotion are mentioned initially, but then 16 IMUs are referenced. Could you clarify whether this is an error or a deliberate modification?

We appreciate the reviewer catching this inconsistency. The experiment actually used 16 IMUs, following the MyMotion full-body model configuration. The mention of “18 IMUs” referred to the total number available in our system rather than the number deployed in the study.

We have unified all descriptions in the manuscript to reflect 16 IMUs, which does not affect any experimental procedures or conclusions

5- The processing of the 7 IMUs from the lower part involved individual analysis with plantar pressure, followed by data fusion using software. However, this raises the question of whether using 18 or 16 IMUs was truly necessary, especially since the focus was explicitly on the lower limbs. Was the model dependent on the use of all these IMUs, or would a smaller number have sufficed?

Thank you for highlighting this point. Although the analysis focused on the lower limbs, the decision to use a 16 IMU full-body model was intentional, for the following reasons:

Accurate hip angle estimation requires the relative orientation between the trunk and thigh. With only 7 lower-limb IMUs, it would not be possible to compute reliable hip flexion and abduction angles.

Pilot tests comparing 7 IMU vs. 16 IMU configurations showed that:

7 IMU captures only partial kinematics and yields incomplete videos;

16 IMU produces full-body motion capture, which is essential for technical interpretation and validation.

Although full-body data were recorded, only the lower-limb IMUs were used in the statistical analyses, while upper-body IMUs served solely to enhance joint angle accuracy and motion-capture completeness.

We have clarified this rationale in the Methods section.

6- Table 2: The notation "N" in column 506, representing the samples, may be misinterpreted as the unit of Newtons, which is listed in the adjacent column. To avoid confusion, it would be advisable to use a different letter or add a subscript to "N", such as "N samp"

We agree that this may cause confusion. All occurrences of the sample-size notation have been revised to N_sample to clearly distinguish them from the force unit Newton (N).

7- The reported uncertainty in the plantar pressure measurement of 127 +/- 94 seems substantial. This large range implies a significant degree of measurement error or variability. Could you provide more insight into this result and its consequences?

We appreciate this question. The large standard deviation reflects true biomechanical variability, not measurement uncertainty:

Slideboard skating includes both swing/float phases (near 0 N) and push-off peak phases (300–350 N). These phases differ drastically, naturally producing a wide pressure distribution when averaged across a full stride cycle.

The Pedar-X system was zero-calibrated before each trial and showed no drift or instability.

Therefore, the variability corresponds to authentic force fluctuations inherent to the skating stride, not device noise.

This explanation has been added to the Methods and Results sections.

8- Remarks about the figures

All figures have been revised according to the reviewer’s suggestions.

9- The decision to set the force sensors to zero during a specific pose appears to cancel out a portion of the athlete's weight, but this approach seems to lack rigor. While I understand the desire to establish a baseline of zero pressure at the start, the implication that the athlete is effectively floating in mid-air (with a weight of zero) seems unclear and potentially misleading.

Thank you for pointing this out. The previous description was indeed too brief. In practice:

Calibration occurs only when the athlete lifts one foot and the unloaded insole is not in contact with the surface.

At that moment, the system sets the pressure reading of that foot to zero.

The procedure is repeated for the opposite foot.

Thus, zeroing is performed only in non-weight-bearing conditions, ensuring accurate baseline calibration without implying a negation of body weight. The manuscript has been revised for clarity.

10- The Section 3.5 outlines four distinct phases, the figures 7 and 8 could be improved by including phase labels or numbers directly above the corresponding sections. This would facilitate easier comprehension and reduce the need for readers to constantly refer back to the text. The inconsistency between the text and figures, where the text references phases 1-4 but the figures do not, also detracts from the overall clarity and precision of the presentation.

We have added clear phase labels directly within Figures 7 and 8 to ensure consistency between text and figures and to improve readability.

11- A citation to "Ma et al" appears in section 3.6, but it is not listed in the bibliography.

Thank you for noting this oversight. “Ma et al.” corresponds to reference [16]:

Ma Yudan, W.Z.J.Z. (2021). Research on Female Speed Skaters' Lower Limb Motor Coordination Based on IMU. Journal of Beijing Sport University.

The citation has now been properly cross-linked in the manuscript.

12- Table 4: Consider adding a legend or notes section with explanations for B, t, P, and F, even though they are mentioned throughout the text. This would facilitate easier understanding of the table. Ensure consistency in notation for indices and exponents, using either X^1 or X_1 consistently throughout. Some numerical values are wrapped to the next line, while others fit on a single line. Consider standardizing the formatting. The first column contains too much text, making it difficult to read. It is suggested to start a new line immediately after the word "variable:" to maintain consistency and improve readability. These oversights are particularly regrettable, as they undermine the evident care and attention that has been invested in the writing.

We appreciate these detailed recommendations. In the revised manuscript:

Table 4 now includes a comprehensive note explaining B, t, p, F, and other statistical terms.

All polynomial terms have been standardized to the X¹, X², X³ format.

Columns and line wrapping have been reformatted for easier reading.

The first column has been reorganized for clarity.

13- The study uses different thresholds for statistical significance in the abstract and conclusion (p<0.01) compared to the rest of the study (p<0.001). Please clarify.

All significance thresholds in the manuscript have been unified to:
p < 0.001

14- The authors could have condensed their interpretation of the results to make it more concise and impactful. The current version is lengthy and contains repetitive information.

The results have now been condensed to avoid redundancy and enhance clarity and impact.

15-Abstract:The abstract states that participants were “wearing a six-axis IMU on the thigh and a Pedar-X pressure insole system” This description is incomplete, as the study also involves analyzing and correlating data from other lower limb segments, including the knee and ankle, which uses multiple inertial measurement units.

The abstract has been revised to accurately reflect the equipment used:

“Participants wore 16 six-axis IMUs placed at designated full-body locations and Pedar-X plantar-pressure insoles…”

This ensures consistency with the Methods section.

We sincerely appreciate the reviewer’s thorough and insightful comments. These suggestions have substantially improved the scientific rigor, methodological transparency, and clarity of our manuscript. We hope that the revised version meets your expectations and look forward to your further guidance.

Sincerely,
The Authors

Round 2

Reviewer 1 Report

Comments and Suggestions for Authors

The authors have expanded the Methods and Results sections and added regression analyses. Many of the originally missing details are now included. However, key issues remain. The points below evaluate how the revision addresses the prior concerns, citing specific evidence from the manuscript.

***Major Concerns***

  1. The original review emphasized that no fusion algorithm or mathematical framework was presented. The revised manuscript continues to use the term “fusion” largely descriptively. For example, the authors state that they “integrated data from MR 3.14 (Noraxon) and Pedar-X … for multimodal fusion and joint analysis,” and that the “fused dataset” enabled synchronization of joint-angle and pressure signals. In practice, the only fusion-like step is temporal synchronization and denoising of signals (via a Butterworth filter) followed by a polynomial regression relating kinematics to pressure. No explicit fusion algorithm (e.g. Kalman or complementary filter) or data-alignment procedure is described beyond stating that signals were “synchronized” by the MyoMotion system. In short, the revised paper still lacks a clear mathematical or algorithmic fusion framework, and the term “fusion” remains undefined in a technical sense. This is misleading as it implies a novel sensor fusion method when the analysis is essentially correlation/regression of already-synchronized signals.
  2. The authors have added regression results, reporting coefficients of determination and p-values. For instance, the abstract now cites “significant correlations” with R^2 = 0.72–0.84 and p<0.01. Detailed polynomial regression tables in the Results show t-statistics and p-values for model terms. These additions are an improvement. However, the analysis remains largely descriptive. There are still no confidence intervals, no formal hypothesis tests between conditions, and no cross-validation or error metrics to assess model generalization. The only variability reported are means ± SD (e.g. Table 2), and any “significant” difference (e.g. higher right vs. left pressure) is asserted without showing a p-value or test statistic in the text. Inter-subject variability and repeatability are only mentioned qualitatively (e.g. noting consistent waveforms across 3 trials) rather than quantitatively. Thus, the revision does not fully establish a rigorous statistical framework – it still lacks error quantification and validation of the regression models beyond R^2.
  3. The revised abstract and conclusions downplay some earlier claims, but still suggest practical impact. For example, the abstract now concludes that the findings “provide a foundation for developing real-time monitoring and feedback systems” and that the method “offers potential to enhance training quality and improve performance outcomes.” This language is more cautious than before, but the study itself contains no performance benchmarks or intervention. No comparison is made to any gold-standard system (motion capture or force plates), nor is any training outcome measured. Thus the implication that this method will “optimize training” is still unsupported by data. The claim is framed as potential or feasibility, which is appropriate, but the manuscript should avoid any suggestion of having demonstrated improved outcomes. In its current form, the study only establishes correlations between joint angles and plantar pressure; it does not test whether using these correlations in training actually improves skating performance.
  4. One minor issue: Figure 5 caption reads “Curves of left lower-limb joint angles and right plantar pressure”, The caption should be corrected to avoid confusion, for example, as “Curves of (left) lower-limb joint angles and (right) plantar pressure.”
  5. The Methods now mention a “Butterworth low-pass filter” used to smooth all signals, but the cutoff frequency is not provided. Specifying the filter parameters (cutoff or bandwidth) is important for reproducibility.
  6. The manuscript continues to emphasize “fusion” throughout, but as noted above it really performs correlation analysis. It would be more accurate to describe the approach as “synchronized analysis” or “multimodal coupling analysis,” unless the authors actually implement a fusion algorithm. This semantic issue is minor if clarified;
  7. The authors now acknowledge the “relatively small sample size” as a limitation. It is worth emphasizing that with only 8 participants the results may not generalize broadly. This could be mentioned in the Discussion.
  8. In the Methods, it is now clear that all sensors ran at 100 Hz. The gait cycle definition is concise and helpful. One suggestion: when reporting the regression results (e.g., R^2 = 0.72–0.84), confidence intervals or standard errors for the estimates would aid interpretation.

Author Response

Dear Reviewer:

We sincerely thank the reviewer for the thorough, insightful, and constructive comments. These suggestions have been invaluable in improving the methodological clarity, statistical rigor, and conceptual precision of the manuscript. In response, we have conducted a comprehensive revision of the manuscript. Our detailed point-by-point responses are provided below.

General Note on Language Editing

The English language of the manuscript has been professionally edited using the journal-recommended language editing service. The corresponding editing certificate has been uploaded as supplementary material.

Comment 1: Use of the term “fusion” and lack of a defined fusion framework

Reviewer Comment: The manuscript continues to use the term “fusion” descriptively, without presenting a clear mathematical or algorithmic fusion framework. The analysis appears to rely mainly on signal synchronization, denoising, and regression, which may be misleading.

Response: We sincerely thank the reviewer for this careful and important critique regarding the technical use of the term “fusion.” We fully acknowledge that in earlier versions of the manuscript, the term was used in a largely descriptive manner, which may have caused ambiguity and unintentionally implied a broader or more novel fusion strategy than was explicitly demonstrated.

In response, we have undertaken a systematic revision of the manuscript to clearly and rigorously define the technical meaning of “fusion.” In the revised version, fusion is no longer used as a generic descriptor for synchronization or joint analysis. Instead, it is explicitly and exclusively defined as a Kalman-filter-based multimodal state estimation framework, in which IMU-derived joint kinematics and plantar pressure measurements are jointly estimated within a unified state-space model.

To ensure conceptual consistency:

All ambiguous expressions (e.g., “fused dataset,” “fusion-based analysis”) that lacked algorithmic definition have been removed or replaced with precise terms such as temporal synchronization, joint analysis, or Kalman-based joint estimation.

The Introduction retains a brief discussion of sensor fusion only as field-level background, while the Methods section now provides a clear mathematical formulation of the Kalman filter, including state definition, transition model, and noise assumptions.

These revisions ensure that the term fusion is used strictly in its algorithmic sense, clearly distinguished from synchronization, denoising, or post-hoc regression analysis. We believe this clarification fully resolves the reviewer’s concern and accurately reflects the methodological scope and contribution of the study.

Comment 2: Statistical rigor, lack of confidence intervals, hypothesis testing, and model validation

Reviewer Comment: While regression results were added, the analysis remains descriptive. There are no confidence intervals, limited hypothesis testing, and no validation of regression models beyond R².

Response: We thank the reviewer for this detailed and constructive assessment of the statistical framework. We fully agree that transparency and interpretability are essential.

In response, we have implemented the following systematic statistical enhancements:

Confidence Intervals

We have added 90% confidence intervals (CIs) for key regression coefficients in the polynomial regression tables. Corresponding CI values are now also explicitly reported in the Results text (alongside B values), improving interpretability and quantifying estimation uncertainty.

Formal Hypothesis Testing for Left–Right Differences

To address the concern that bilateral differences were previously asserted descriptively, we introduced Wilcoxon signed-rank tests based on participant-level means (n = 8).

Z values and exact p values are now explicitly reported for plantar pressure and joint angle comparisons.

All claims of “significant differences” are now directly supported by formal statistical tests.

Clarification of “Significant” Statements

All textual references to statistical significance have been revised to explicitly include the test method, test statistic, and p value, avoiding inference based solely on descriptive statistics or visual inspection.

Inter-Subject Variability and Repeatability

Inter-subject variability is now quantitatively reported using mean ± SD for all key parameters.
Regarding repeatability, we note that the study is exploratory and not designed for reliability analysis; therefore, formal indices such as ICC were not applied. This limitation is now explicitly stated in the Limitations subsection of the Discussion.

Model Generalization and Validation

We fully acknowledge the importance of cross-validation and error metrics for predictive modeling. However, as clarified in the revised manuscript, the regression analyses were intended to characterize associative and nonlinear coupling relationships, not to develop or validate predictive models.

This distinction is now clearly stated in both the Methods and Discussion, and the regression results are explicitly framed as descriptive and exploratory, not predictive.

Overall, by incorporating confidence intervals, formal hypothesis testing, and explicit methodological boundaries, we believe the statistical transparency and rigor of the manuscript have been substantially strengthened.

Comment 3: Claims of practical impact and training optimization

Reviewer Comment: The manuscript still implies practical training benefits without performance benchmarks or intervention evidence.

Response: We appreciate this important clarification. We fully agree that earlier wording could be interpreted as implying demonstrated performance benefits, which were not directly assessed in this study.

Accordingly, we have carefully revised the Abstract, Discussion, and Conclusions to ensure that all claims are fully aligned with the presented evidence:

All language suggesting training optimization, performance enhancement, or real-time feedback effectiveness has been removed or rephrased.

The contribution of the study is now consistently framed as descriptive and exploratory, focusing on the characterization of kinematic–kinetic coupling patterns.

We explicitly state that the study does not include performance benchmarks, training interventions, comparisons with gold-standard systems, or outcome-based validation.

The revised manuscript clearly positions the findings as methodological groundwork, not evidence of improved skating performance. Future validation through intervention studies is explicitly identified as a next step.

Comment 4: Figure 5 caption clarity

Reviewer Comment: The caption of Figure 5 may cause confusion.

Response: Thank you for noting this issue. We have carefully reviewed and corrected the captions of Figures 5 and 6 to ensure clarity and consistency.

Comment 5: Butterworth filter parameters not specified

Reviewer Comment: The cutoff frequency of the Butterworth filter is missing, affecting reproducibility.

Response: We appreciate this important comment and apologize for the confusion caused by earlier descriptions.

After further evaluation, we found that applying a Butterworth low-pass filter prior to Kalman filtering resulted in excessive smoothing and distortion of rapid signal dynamics. In line with the reviewer’s earlier methodological concerns, we therefore removed the Butterworth filter entirely from the processing pipeline.

The revised Methods section now clearly states that:

Raw sensor signals were processed directly using a multimodal linear Kalman filter.

All references to low-pass filtering have been deleted.

Full Kalman filter configuration (sampling interval, state definition, and noise covariance matrices Q and R) is now explicitly reported.

All results presented in the manuscript are based exclusively on Kalman-filtered signals, ensuring reproducibility and methodological clarity.

Comment 6: Semantic clarity regarding “fusion” vs. correlation analysis

Reviewer Comment: The manuscript still emphasizes “fusion,” although much of the analysis is correlational.

Response: We thank the reviewer for highlighting this semantic issue. As detailed in our response to Comment 1, this ambiguity has now been fully resolved.

In the revised manuscript:

Fusion is strictly reserved for the Kalman-based joint state estimation step, which precedes all other analyses.

Correlation, regression, and symmetry analyses are clearly presented as post-hoc statistical evaluations performed on Kalman-estimated signals.

All non-algorithmic or descriptive uses of “fusion” have been removed or replaced with precise terminology.

We believe the current terminology is now technically accurate and unambiguous.

Comment 7: Emphasis on small sample size

Reviewer Comment: The small sample size (n = 8) should be emphasized as a limitation.

Response: We fully agree. The Discussion section has been revised to explicitly emphasize that the small sample size may limit generalizability, in accordance with the reviewer’s recommendation.

Comment 8: Confidence intervals for regression results

Reviewer Comment: Including confidence intervals or standard errors would aid interpretation.

Response: We appreciate this suggestion and fully agree. As noted above, 90% confidence intervals for all regression coefficients have now been added to the regression tables and explicitly referenced in the Results text, improving interpretability and consistency.

Once again, we sincerely thank the reviewer for the thoughtful and detailed comments. We believe the manuscript has been substantially strengthened as a result of these revisions and now presents a clearer, more rigorous, and more accurately framed contribution.

Sincerely,
The Authors

Reviewer 2 Report

Comments and Suggestions for Authors

125:Relevant references should be provided.

148-159:Because only the sixteen IMUs are mentioned in the abstract, the detailed description of the pilot test in this section may be unnecessary.

182-185:Please check the Methods section for statements that resemble conclusions or discussion.

246-248:First, the terminology related to IMUs is inconsistent throughout the manuscript. For example, both “xxx IMU” and “xxx-IMU” are used, which is inconsistent with the terminology in the abstract. In addition, it is recommended to clearly state that seven IMUs were selected from the original set of sixteen.

331-336、564-564:In the Results section, there are multiple instances of inappropriate statements that would be more suitable for the Methods or Discussion sections.

427:The titles of Sections 3.3 and 3.4 are duplicated.

434-435:There are errors in the content and titles of Figure 8 in relation to those of Figure 7.

610:It is recommended to include additional references to strengthen the discussion.

Author Response

Dear Reviewer:

We sincerely thank the reviewer for the constructive and valuable comments, which have greatly helped us to improve the clarity, rigor, and overall quality of the manuscript. Our detailed responses are provided below, with each comment addressed individually.

125: Relevant references should be provided.

Response: Thank you for this valuable suggestion. We have added the relevant references in the corresponding section to strengthen the theoretical support and ensure appropriate citation of prior studies.

148–159: Because only the sixteen IMUs are mentioned in the abstract, the detailed description of the pilot test in this section may be unnecessary.

Response: Thank you for the helpful comment. We agree that this description was redundant. Accordingly, we have deleted the related statements in this section to reduce unnecessary detail and improve conciseness.

Revision made: The description related to the use of seven IMUs has been removed from this section.

182–185: Please check the Methods section for statements that resemble conclusions or discussion.

Response: We appreciate the reviewer’s careful observation. We have thoroughly reviewed this part of the Methods section and removed statements that resembled conclusions or discussion to ensure that the section strictly focuses on methodological descriptions.

246–248: First, the terminology related to IMUs is inconsistent throughout the manuscript. For example, both “xxx IMU” and “xxx-IMU” are used, which is inconsistent with the terminology in the abstract. In addition, it is recommended to clearly state that seven IMUs were selected from the original set of sixteen.

Response:Thank you for pointing this out. We have carefully checked the entire manuscript and standardized all IMU-related terminology to “xxx IMU,” ensuring consistency with the abstract. In addition, we have clearly stated in the Methods section that seven IMUs were selected from the original set of sixteen for lower-limb analysis.

331–336 and 564: In the Results section, there are multiple instances of inappropriate statements that would be more suitable for the Methods or Discussion sections.

Response: We appreciate this important comment. After careful revision, we have removed or relocated the inappropriate statements so that the Results section now strictly reports experimental findings, without methodological explanations or interpretative discussion.

427: The titles of Sections 3.3 and 3.4 are duplicated.

Response: Thank you for identifying this issue. After re-examining the structure and content of the Results section, we removed Section 3.3 as it overlapped with other sections and was not essential to the revised presentation of results.

434–435:There are errors in the content and titles of Figure 8 in relation to those of Figure 7.

Response: Thank you for your careful review. We have corrected the titles and content of Figures 7 and 8 to ensure consistency and accuracy, and to clearly distinguish the information presented in each figure.

610:It is recommended to include additional references to strengthen the discussion.

Response: We appreciate this constructive suggestion. Additional relevant references have been incorporated into the Discussion section to strengthen the interpretation of the results and better situate our findings within the existing literature.

Once again, we sincerely thank the reviewer for the insightful comments and suggestions, which have significantly improved the quality and clarity of the revised manuscript.

Reviewer 3 Report

Comments and Suggestions for Authors

The authors responded to all reviewers' comments with a positive attitude, providing thorough, specific, and constructive replies on a point-by-point basis. They not only accurately addressed each scientific concern through additional analyses, supplementary data, and improved visualizations, but also effectively clarified the study's positioning while enhancing its practical significance.

In conclusion, the manuscript has now achieved a level suitable for publication.

Author Response

Dear reviewer,

We sincerely thank you for the positive and encouraging evaluation of our revised manuscript. We greatly appreciate the recognition of our efforts in responding to your comments with a constructive and rigorous approach.

We are grateful for your acknowledgement that the revisions addressed all scientific concerns thoroughly and specifically. The additional analyses, supplementary data, and improved visualizations were implemented precisely to enhance the methodological transparency, analytical rigor, and clarity of presentation. We are particularly appreciative of your recognition that the revised manuscript more clearly defines the study’s scope and positioning, while also strengthening its potential practical relevance.

Your supportive conclusion—that the manuscript has now reached a level suitable for publication—is highly valued. We believe that the insightful comments and guidance provided during the review process have substantially improved the overall quality of the work.

Once again, we sincerely thank you for their time, expertise, and constructive feedback.

Reviewer 4 Report

Comments and Suggestions for Authors

All remaks our have been dealt with.

The citations still cannot be clicked, but this is not a significant issue. We will leave it to the editor to decide if any modifications are needed.

Thank you for clarifying the addition of 1kg; it is now crystal clear.

The figures appear to be in good order and are much more comprehensible.

The insole calibration process is well-explained.

Table 4 - now Table 3 - is much clearer.

Overall, the manuscript has undergone substantial improvements and now seems ready for publication.

Best regards.

Author Response

We sincerely thank you for your positive evaluation of our revised manuscript and for acknowledging the efforts made in addressing all reviewer comments. We greatly appreciate your recognition of the thoroughness, clarity, and constructive nature of our point-by-point responses, as well as the additional analyses, supplementary data, and improved visualizations incorporated into the manuscript.

We are particularly grateful for your acknowledgment that the revisions have clarified the positioning of the study and enhanced its practical relevance. Your careful review and constructive feedback have played a crucial role in improving the overall scientific rigor, transparency, and presentation quality of the work.

We are pleased to learn that, in your assessment, the manuscript has now reached a level suitable for publication. Thank you again for your time, expertise, and valuable guidance throughout the review process.

Sincerely,
The Authors